# Local generation and efficient evaluation of numerous drug combinations in a single sample

**Vlad Elgart[1,2], Joseph Loscalzo[1,2]***

[1]Department of Medicine, Brigham and Women's Hospital, Boston, United States; [2]Harvard Medical School, Boston, United States

**Abstract** We develop a method that allows one to test a large number of drug combinations in a single-cell culture sample. We rely on the randomness of drug uptake in individual cells as a tool to create and encode drug treatment regimens. A single sample containing thousands of cells is treated with a combination of fluorescently barcoded drugs. We create independent transient drug gradients across the cell culture sample to produce heterogeneous *local* drug combinations. After the incubation period, the ensuing phenotype and corresponding drug barcodes for each cell are recorded. We use these data for statistical prediction of the treatment response to the drugs in a macroscopic population of cells. To further application of this technology, we developed a fluorescent barcoding method that does not require any chemical drug(s) modifications. We also developed segmentation-free image analysis capable of handling large optical fields containing thousands of cells in the sample, even in confluent growth condition. The technology necessary to execute our method is readily available in most biological laboratories, does not require robotic or microfluidic devices, and dramatically reduces resource needs and resulting costs of the traditional high-throughput studies.

**\*For correspondence:** jloscalzo@rics.bwh.harvard.edu

## Editor's evaluation

This paper provides a road map for evaluating drug combinations using a single sample. It should be particularly useful for other biologic laboratories to test agents in a manner that does not require major resources, or microfluidics and could reduce costs and time to required for ultimate testing in vivo.

## Introduction

The optimization of targeted therapies is based on the determination of drug dose-response and the selection of a regimen that is best suited for an individual patient. A critical barrier to optimization screening is the availability and amount of relevant tissue sample (e.g. biopsy), especially in the case of combination therapy. This limitation is a reflection of the fact that the number of possible drug combinations grows exponentially with the number of drugs and dosages (cf., Supporting Information). For example, the number of all possible combinations of three drugs tested at ten different concentrations (including zero) is $10^3$. This number does not include the biological replicates and controls that render the careful testing of even three drugs extremely challenging.

There exists another challenging aspect to the identification of optimal drug dosage and regimen: accurate reproducibility of in vivo and in vitro experiments is notoriously difficult to achieve in many biological applications, such as drug dose-response quantification. Many confounding factors contribute to the variability in sample readout *Balázsi et al., 2011*; *Niepel et al., 2019*; these factors

can generally be grouped into two categories, which are usually described as intrinsic and extrinsic noise.

Originally, this noise nomenclature was used to describe temporal fluctuations in a single protein within a single cell (*Elowitz et al., 2002*; *Raj and van Oudenaarden, 2008*). Here, we adopted this classification to an autonomous system which is an ensemble of cells isolated from other samples, but still subject to fluctuations in the external environment. Instead of measuring temporal fluctuations in individual cells, one can record snapshot(s) of cellular phenotype(s) within a single sample. The source of noise in these measurements is variability between individual cells within the sample; we refer to this type of noise as intrinsic. Noise that is driven by external factors to this autonomous system (a particular sample) is referred to as extrinsic. Variability between different samples due to temperature, humidity, etc., are examples of extrinsic noise in this context.

Sources of intrinsic noise in drug uptake by and action within cells include fundamental biochemical processes, such as passive uptake, active transport, and degradation. By contrast, external factors that contribute to noise are principally manifest through convection and advection. Namely, these external factors are related to movement and gradient formation of fluid or gas. The initial cell seeding that eventually leads to unique microenvironment formation can also be thought as an example of extrinsic noise using this terminology.

While intrinsic cell variability can be significant, we believe that it is the extrinsic factor(s) that drive sample variability in most experimental cellular systems. This conclusion derives from the Law of Large Numbers (*Feller, 1968*), since the typical number of cells in a culture sample is at least of the order of $10^4$, while the number of biological/technical replicates is ~3.

If the observed sample variability is due to extrinsic noise then it is very tempting to conduct all drug response experiments in the same tissue or cell culture sample. Indeed, under these conditions, it is guaranteed that external fluctuations in the environment are exactly the same for all testing conditions. Hence, quantification of the drug's response does not require averaging over many different samples to achieve an accurate, *relative* comparison of the drug's effectiveness.

In order to test different drug combinations in the same cell culture or tissue specimen, one must create these combinations *locally*. One of the smallest possible quantifiable locales in both cell culture and tissue samples is a single cell. Indeed, we can treat each cell as an individual chemical "reactor" in which testing of different drug combinations is performed. Thus, if we can (i) create different drug combinations in individual cells, and (ii) subsequently quantify these combinations and the ensuing outcome (phenotype) for each cell, the logistics of testing combinations in a single macroscopic sample would be effectively addressed.

There exists a straightforward way to resolve the quantification issue (ii) above. Drugs can be uniquely labeled (e.g. using fluorescent tags), in which case, the amount of drug taken up by each cell would scale as the fluorescence of its corresponding tag.

Let us now address a more delicate problem (i): how to create a broad range of drug combinations in different individual cells. At first glance, an herculean effort would be required to accomplish this task. Fortunately, cells themselves solve this problem effortlessly for us. Even cells with identical DNA typically exhibit deviations in drug uptake. Therefore, the random nature of drug(s) uptake by individual cells can be utilized to create different drug combinations. Here, we further enhance heterogeneity and independence in *stochastic* drugs mixing by creating individual drug gradients across the sample.

Rather than delivering drugs homogeneously, we rely on simple forced drug advection (by local injection) to create transient local drug gradients independently for each drug. After delivery and incubation of random *local* drug combinations to individual cells, we analyze the ensuing cellular phenotypes and corresponding drug optical barcodes by fluorescence microscopy or flow cytometry.

Cell 'barcoding' terminology in biology usually implies DNA identification, but here we use fluorescent signal(s) instead to encode drug regimen. This method was used by Krutzik and Nolan to distinguish between individual *samples* exposed to different drug regimens for high-throughput analysis (*Krutzik and Nolan, 2006*). The number of samples in their experiments was equal to the number of drug conditions, since the authors were barcoding samples, not individual cells.

The statistical analysis of the imaging data is used to predict the effect of ratiometric dosing on the outcome (cell phenotype). In this fashion, we require as little as a single cell culture sample to test all

feasible drug combinations, bypassing the need for an otherwise enormous number of controls and technical replicates.

In this work, fluorescently labeled drugs are clustered into three 'barcoding' classes distinguished by the physical association of the label with the drug: either irreversibly linked or cleavable/separable chemistry. Additionally, we developed a labeling method that does not require any chemical modification of the drug(s), such as covalent fluorescent tagging. This particular approach involves local co-injection of an unconjugated pair of pre-mixed drug and fluorescent dye that serves as a correlative barcode. We utilize the fact that on short incubation time scales (less than an hour), the initial injection producing forced advection is mainly responsible for transient gradients in the sample. Immediately after local injection, the drug and its tag concentration profiles across the sample correlate. The subsequent diversion of these profiles due to differences in diffusion and transport rates between the drug and its unconjugated tag occurs slowly (*di Cagno and Stein, 2019*) over this comparatively short incubation period, ensuring persisting correlation between drug and tag uptakes throughout the course of the typical experiment. For all drug barcoding classes we designed and tested, we rely on the fact that the drug barcode is retained by the cells. Hence, total drug exposure over time (area under the curve) can be assessed by a single time point measurement.

We also developed segmentation-free microscopy-based image analysis and statistical data processing capable of handling large optical fields containing thousands of cells in the sample, even in confluent growth conditions. These algorithms allow us to bypass the standard segmentation process, typically requiring additional fluorescent marker(s), significant computational time, and resources.

To demonstrate a range of possible applications of our method using different drug classes, we include implementation of our stochastic drug mixing strategy covering different biological topics. We first consider a benchmark system in which 'drugs' are diester dyes with distinguishable fluorescent properties. To go beyond this 'toy' system, we assessed the effect of a combination of siRNAs on a direct and quantifiable target, viz., GFP fluorescence, as modified by siRNA-dependent expression of GFP protein. This system is characterized by a simple phenotype readout, such as an unique fluorescent signal, and the modulation of the phenotype is a result of the direct drug-target interaction, and not a secondary, downstream effect.

Lastly, we conducted multiple cell culture experiments treating cells with combinations of drugs commonly used in multi-drug chemotherapy regimens, such as cisplatin, docetaxel, cyclophosphamide, and 5-fluorouracil. We adopted our technology to accommodate quantification of cell death as a phenotype common to all of these agents.

Our approach has distinct advantages over conventional drug combination screening methods, including far greater efficiency, a smooth (continuous) variation of drug concentrations in the resulting combinations, and concomitant phenotype readout. We believe this study demonstrates the applicability of this method to practical challenges in combination drug development.

## Results

### Stochastic drug mixing

Stochastic mixing occurs naturally due to randomness in the drug(s) uptake process by individual cells. We enhance the random process of drugs' uptake by creating transient drug gradients across the sample.

In the simplest possible implementation of *deterministic* gradient mixing, one can create a drug $A$ gradient along the sample's x-axis and a drug $B$ gradient along its y-axis, generating (locally) drug $A$ and drug $B$ combinations in a continuous fashion. This approach, however, requires microfluidic devices that would maintain stable drug gradients, which can be technically challenging (*Wei et al., 2019*; *Kuo et al., 2019*). Moreover, the implementation of multiple (i.e. >2) drug gradients ex vivo is probably not feasible on a small spatial scale. It is also not obvious as to how to prepare efficient combinations of three or more drugs using this approach.

Here, we used a much simpler approach involving local point injection to create spatio-temporal drug gradients in (mammalian) cell culture. We do not homogeneously distribute drugs in the media but, rather, inject them locally into cell culture in separate, discrete locations. Point injection of multiple drugs ensures chaotic/stochastic mixing patterns across sample (*Box 1*, *Figure 1*).

One barcoding option is to label the drug directly with a fluorescent tag either by chemical modification of the drug or other irreversible binding methods. Here, we demonstrate the application of our method for irreversibly tagged drugs using cell-retained dyes as a benchmark drug model. The cell-permeable diester dye, carboxy fluorescein succinimidyl ester, and its derivatives are an example of this type of drug.

We first consider a benchmark system in which 'drugs' are diester dyes with distinguishable fluorescent properties. These dyes are well retained by the cell for an extended period and, hence, can be used to characterize total uptake by individual cells. Typical targets of these 'drugs' are intracellular thiol or amine groups (Materials and methods).

In *Figure 2a*, we demonstrate the typical results of a random drug mixing experiment using a homogeneous delivery method. HeLa cells were exposed to four different dyes mixed together simultaneously in suspension cell culture according to the manufacturer's protocol and imaged using confocal microscopy. The color of each dye (violet, green, yellow, and red) corresponds to its specific fluorescence emission maximum. The color and intensity of each cell in the image correspond to a unique 'drug' combination.

One reason as to why the majority of cells exhibit somewhat similar color is the high degree of similarity in chemical and, hence, transport properties of these compounds. There is a propensity for cells to take up similar molecules proportionally, with only atypical cells (outliers) exhibiting different behavior in this case.

We experimented with multiple strategies for increasing variability, such as decreasing incubation time and cell culture volume, factors that we showed previously drove heterogeneity in drug uptake (*Elgart et al., 2018*). We found that the most robust way to control variability in uptake is to generate transient drug gradients in culture. A typical outcome of these experiments with HeLa cells is shown

## Box 1. Barcoding individual cells using stochastic drug mixing One can think of transient drug gradients as local, short incubation processes that facilitate uptake heterogeneity across the sample.

The duration(s) of these local uptake processes are governed by diffusion, advection, and the drug-target reaction kinetics in cell culture.

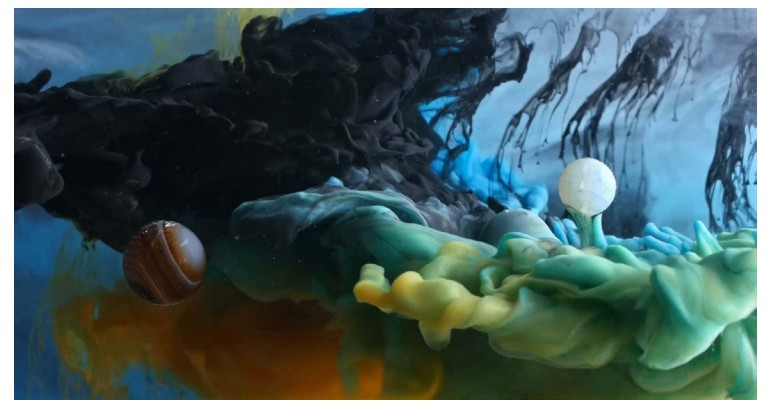

**Box 1—figure 1.** The aesthetic visualization of the stochastic mixing method implemented on a much larger scale. Ink In Motion by Macro Room, https://www.youtube.com/embed/ICxC5ekWnUc. We use the fluorescent intensity of each uniquely labeled drug within a cell as a proxy to corresponding total drug exposure (area under the curve). This allows us to barcode each cell with information about the drug regimen to which it was exposed.

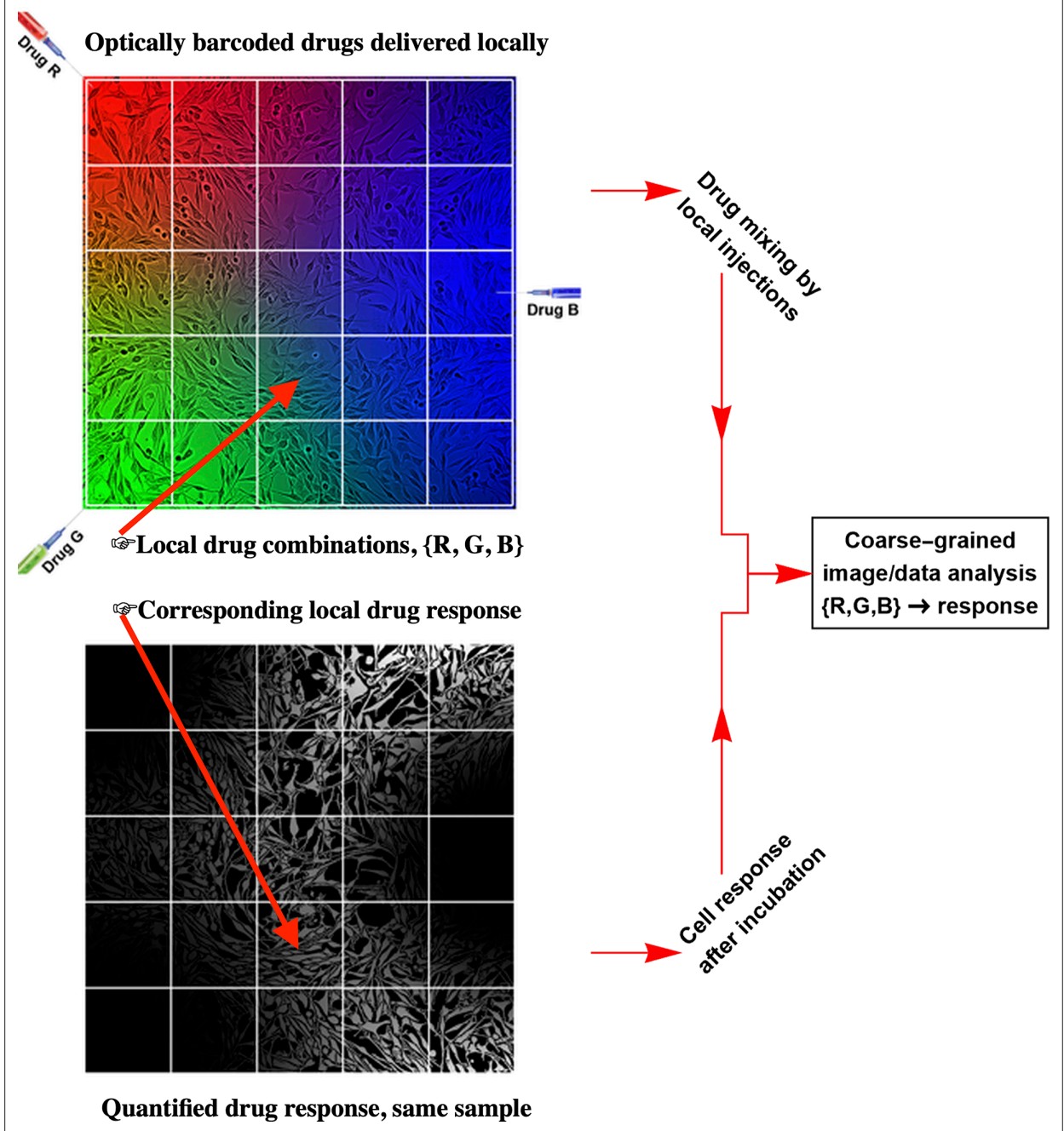

**Figure 1.** The workflow of the temporal local gradient method: Random drug mixing is achieved by creation of the local temporal gradients across a cell culture sample. Cells exposed to different drug combinations are fluorescently barcoded. The drug combination and corresponding outcome (phenotype) are analysed using coarse graining approach to reduce biological and technical noise.

in *Figure 2b* as a raw image. Here, we used a rather crude and simple method to establish transient spatial gradients in suspension cell culture by point injection (Methods).

The quantitative comparison of homogeneous and point injection-based dye delivery in terms of the fluorescence intensity distribution and pairwise correlation relationship is shown in *Figure 2c* and *Figure 2d*, respectively. Additional heterogeneity introduced by point injections leads to a widening of the fluorescence intensity distributions and, thus, a significantly broader concentration range compared to the homogeneous mixing experiment.

We next conducted parallel experiments with adherent HeLa cells using the gradient mixing method by point injection of different drugs in different locations in the culture well (Methods).

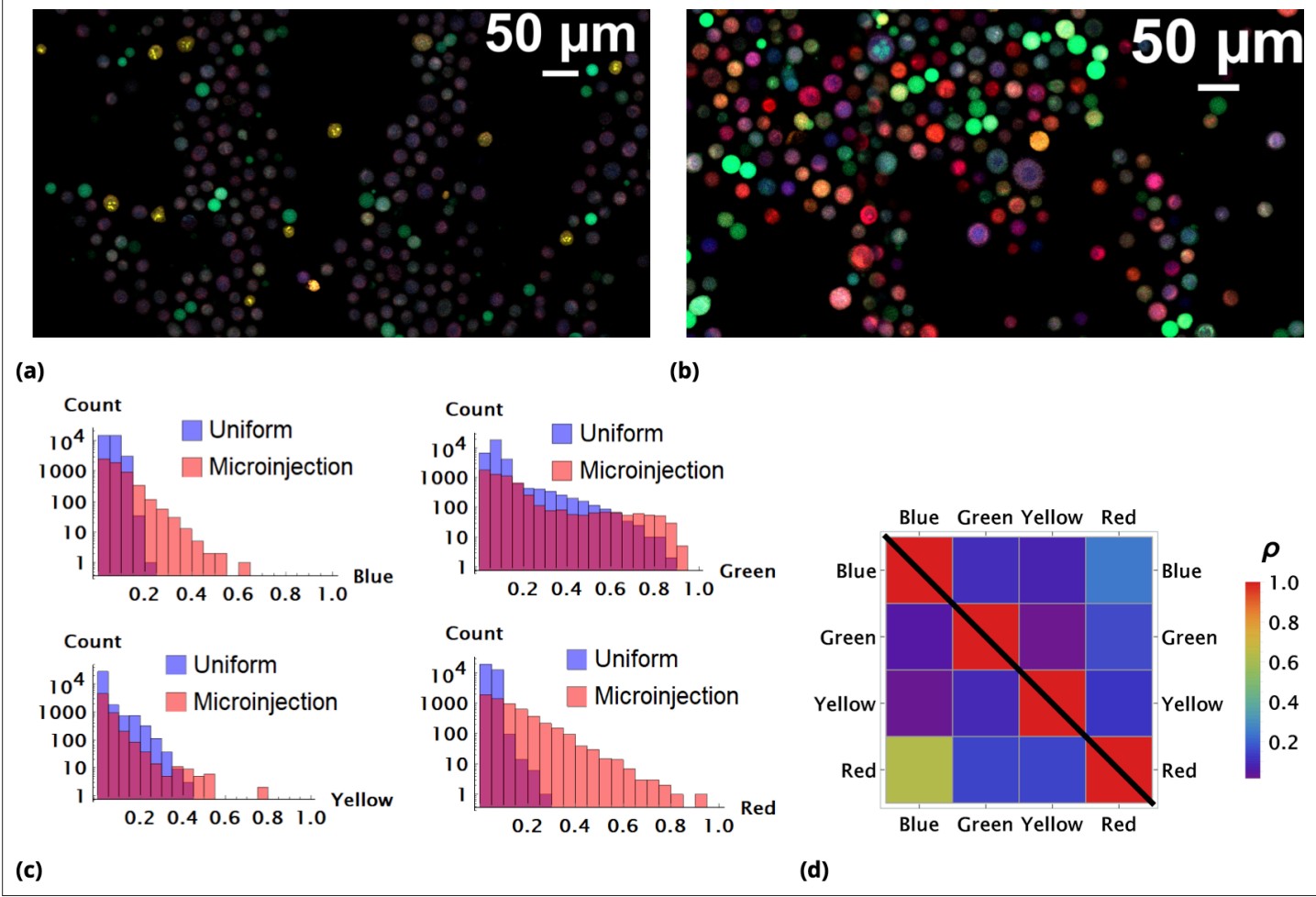

**Figure 2.** Enhancement of random drug mixing in suspension culture. (**a**) and (**b**), Results of random dye mixing in suspension culture; raw image data are shown for homogeneous and point injection delivery methods, respectively. Here, we used four different CellTrace dyes (Violet, CFSE, Yellow, and Far Red), labeling independently and sequentially as for dyes $A$ and $B$ in Section 6. (**c**) The uptake distribution of individual CellTrace dyes incubated with HeLa cells in suspension is shown for each of the conditions on the same plot. Here, each histogram displays cell count as a function of fluorescence intensity, measured in arbitrary units for each dye. (**d**) Correlation coefficients, $\rho$, for all pairwise dye combinations are shown graphically for homogeneous and point injection deliveries on lower and upper triangular matrix plots, respectively. (For example, the CV values of $\rho_{21}$ and $\rho_{12}$ correspond to intensity correlation of Violet and CFSE dyes for homogeneous and point injection delivery methods, respectively.) The absolute value of the correlation coefficient is represented by colors varying from violet to red for values 0 and 1, respectively.

Due to the dispersion process, the initial drugs' distribution is spread ('averaged') locally across finite regions of space (spread size depends on drug diffusion and uptake rates, and also incubation time). This observation allows one to simplify the image processing by averaging fluorescence signal intensities not over individual cells, but, rather, over the mesoscopic regions of space containing multiple cells (Materials and methods). This simplification is critical since confluent cell growth conditions (necessary for a large statistical ensemble) usually present a problem for image segmentation. The statistics obtained using this approach describes typical local cell drug uptake within the given mesoscopic region (tile).

The results of a typical experiment utilizing point injections are compared to homogeneous dye delivery in *Figure 3*. Here, we either co-delivered all dyes homogeneously or locally by point injection. Point injections were performed by local delivery of two dye pairs simultaneously: One pair of *pre-mixed* dyes was codelivered in one point location and another pair of *pre-mixed* dyes was injected at a different point location on the microscopy cell culture slide. As in the case of cells grown in suspension culture, point injection greatly increased the range of drug uptake across the population compared to homogeneous delivery (*Figure 3a*).

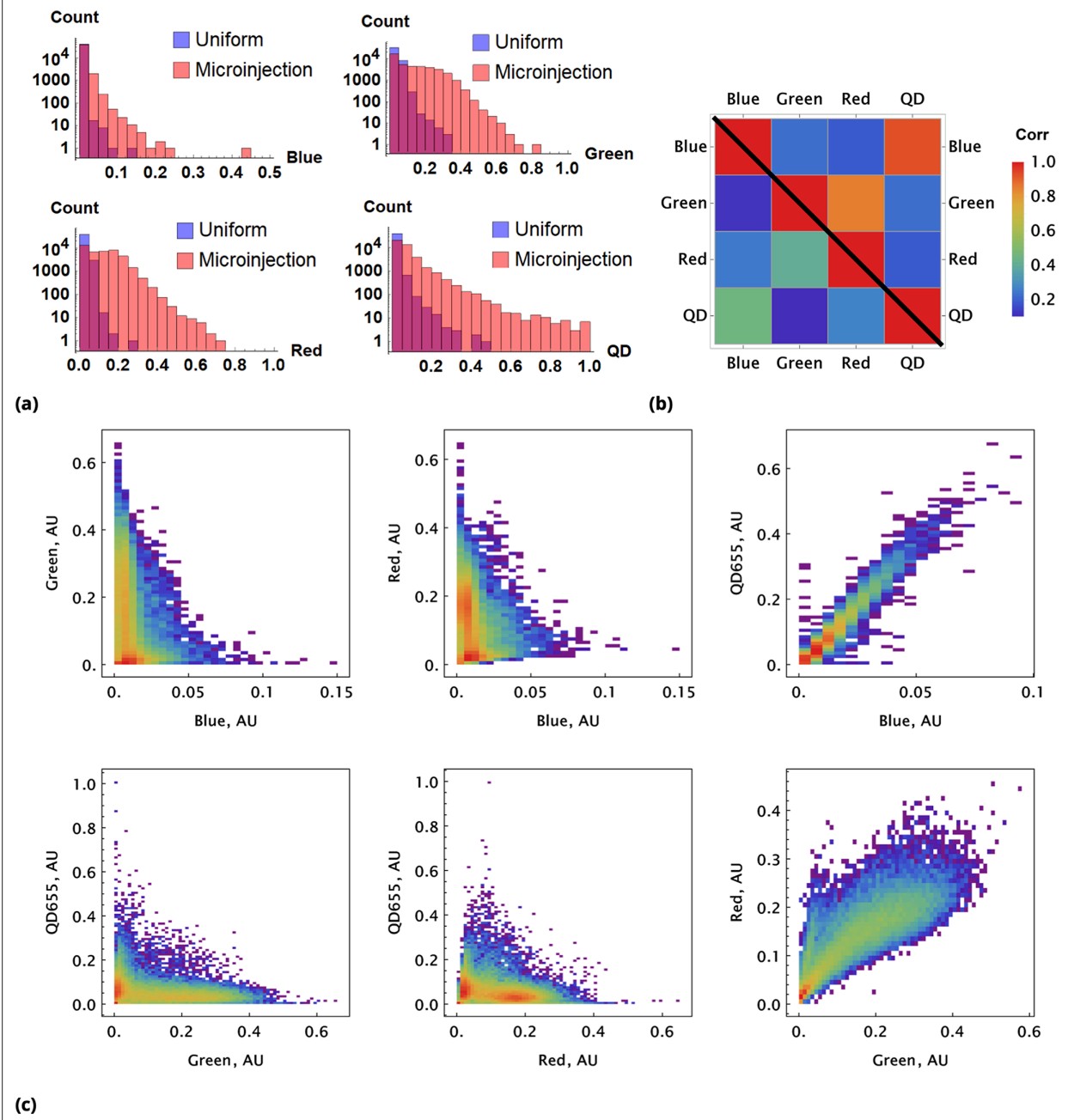

**Figure 3.** A pair of Green and Red CellTrace dyes was co-delivered in one point location and another pair of Blue and QD$_{655}$ dyes was co-delivered in another point location simultaneously of adherent cell culture sample. (**a**) and (**b**), Statistical properties of homogeneous and point drug injections (microinjection) are compared in the same fashion in Sections 6 and 6. (**c**) Scatter density plots for drug uptakes are shown for each pair of dyes tested. First two columns, independent delivery of each dye pair; last column, co-delivery of each pair.

We observed correlation patterns between dye uptakes that were dependent on whether dyes were paired by point delivery or not. Locally co-delivered dyes (point-injection) exhibited a far greater degree of correlation compared to those in unmatched pairs of drugs delivered to different slide locations (*Figure 3b*). This behavior is apparent from the scatter plots shown in *Figure 3c* where color represents local cell count (higher count corresponds to a redder color in the color spectrum scheme). Dye pairs delivered independently by the point-injection method (two different culture slide locations) led to the generation of different mixing ranges of dyes (first two columns, *Figure 3c*).

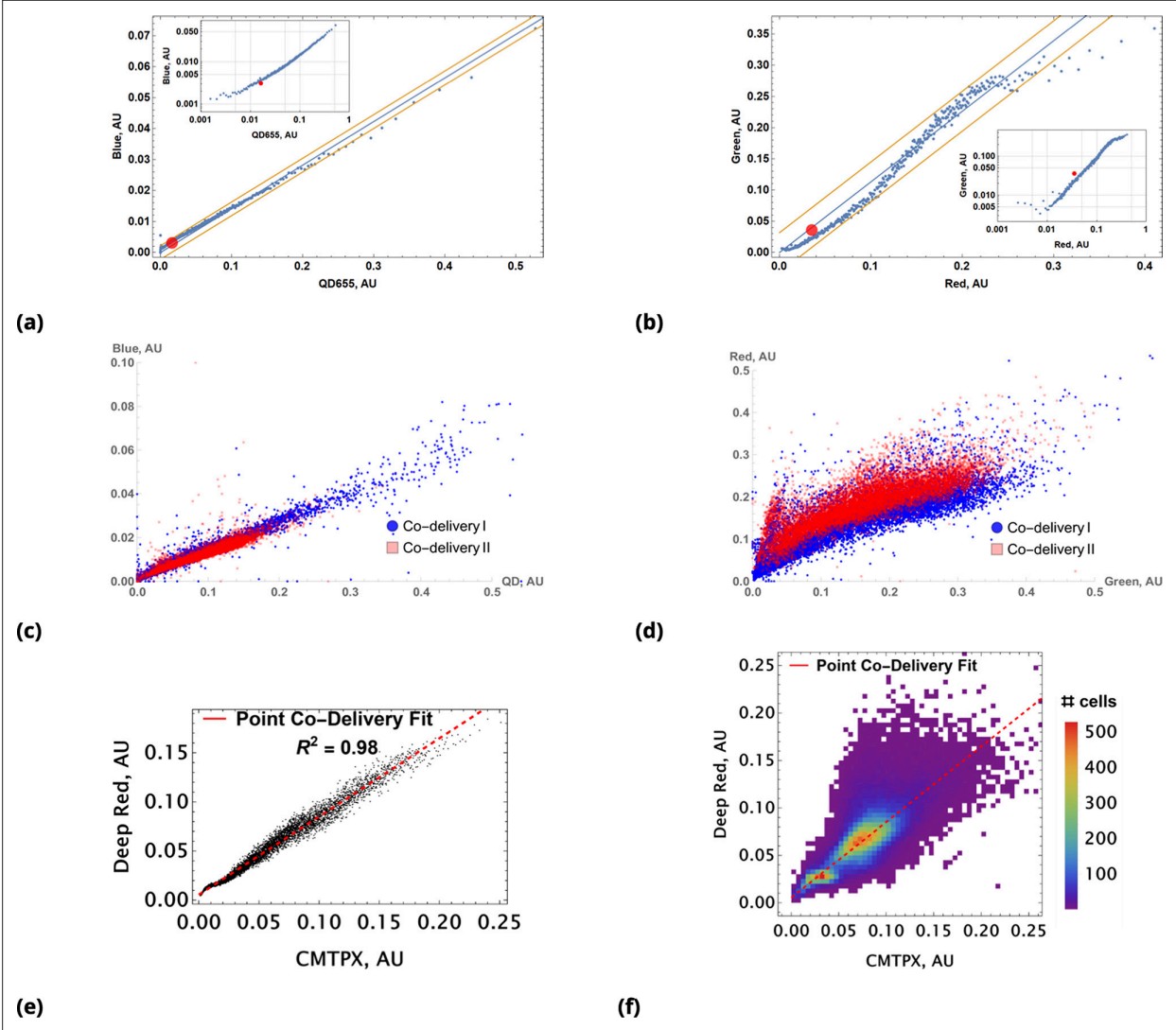

**Figure 4.** Coarse grained scatter plots for pairs of point co-delivered drugs are shown for (**a**) {Blue, QD$_{655}$} and (**b**) {Green, Red} dyes. Log-log scale of dye intensities are shown in insets. Here, the large, red-filled circle on each of the plots corresponds to average values of dye intensities for the homogeneous delivery method. The data from two different point injection experiments are compared for pairs {Blue, QD$_{655}$} and {Green, Red} in (**c**) and (**d**). In these two experiments, the locations of point injections were altered to demonstrate the independence of co-delivered dyes parameterized from the geometric setting. The linear model fits of coarse-grain data are $R^2$ 0.99 and 0.97 for these pairs. (**e, f**) Point co-delivery of CellTracker dye pairs (CMTPX and Deep Red) in fixed cells: (**e**) coarse-grained intensity data and its linear regression fit (**f**) the combined scatter plot of single-cell intensity data for two separate experiments in which dyes were delivered at homogeneous concentrations. Here, we used the same molar ratio of dyes ($1 : 1$) as in the point injection delivery, but delivered dye combinations homogeneously at concentrations of either $5 \, \mu M$ or $10 \, \mu M$. The bi-modality is due to the separation of individual distributions of single-cell intensity data for each of these experiments.

The pre-mixed dyes co-delivered locally exhibit a high degree of correlation with coefficients of correlation greater than 0.8 (cf., right column in *Figure 3c*), even for structurally and chemically different agents such as CellTrace and nanoparticle QD dyes. The underlying cause of this surprising behavior is identical *initial* dispersion of pre-mixed co-delivered dyes throughout a sample by forced advection. The subsequent diversion of concentration profiles due to differences in dye diffusion, transport, and reaction rates occurs slowly over this comparatively short incubation period, ensuring persisting correlation between dye uptakes throughout the course of the typical experiment.

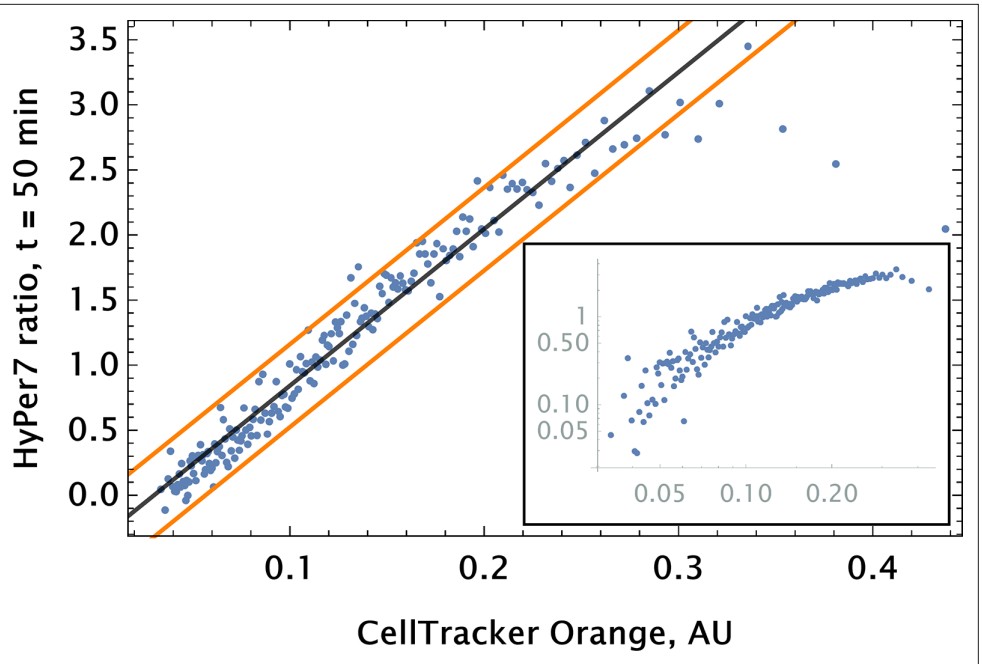

**Figure 5.** Auranofin conjugation-free barcoding in HPAEC: fluorescence signal of barcode (CellTracker Orange) and drug action proxy HyPer7 sensor ($H_2O_2$ fluorescent detector) were imaged by confocal microscopy. Auranofin was pre-mixed with CellTracker Orange dye ($10\,\mu M$ concentration each) and co-delivered locally using point injection. Quantification of the relationship between drug and dye uptake was performed using coarse grained analysis. The linear model fit for the coarse-grained HyPer ratio and 95% confidence intervals are shown by black and orange lines, respectively (adjusted $R^2 = 0.89$). The same scatter data are shown on log-log scale in the inset to demonstrate the broad log-linear range ($\sim 100$ fold) of tag and drug correlation.

## Conjugation-free drug barcoding

One can think of one of the dyes in co-delivered pairs as a drug and another as a label/tag. Since we can predict drug uptake utilizing the high degree of correlation with the corresponding tag, no physical conjugation of drug and tag is necessary.

Here, we rely on the similarity of *temporal* concentration profiles of drug and dye within cell culture. Drug (and dye) uptake is governed by transport and diffusive processes, as well as molecular interactions with its targets, both specific and non-specific (**Elgart et al., 2018**).

In many cases, strong, specific drug binding to the target leads to fast reaction rates which, in turn, result in the uptake process being diffusion/transport-limited. In this situation, media concentration profiles created by the *initial* dispersion drive intracellular drug and dye uptake. We pre-mixed reagents used for co-delivery in a small volume that is injected into a much larger sample volume. Hence, point co-delivery results in similar local concentration profiles of drug and dye, at least for short incubation periods.

Unlike chemically labeled drugs (where the one-to-one correspondence between tag fluorescence and drug uptake is assured by design), the conjugation-free barcoding method requires statistical processing. This step is necessary to filter out the noise due to intrinsic fluctuation in both drug and tag uptakes and improve correspondence between tag fluorescence and drug uptake.

A demonstration of the predictive accuracy in estimating drug uptake using this conjugation-free approach is shown in **Figure 4**. Here, coarse-grained uptakes of co-delivered dye pairs are shown in the scatter plots, **Figure 4a** and **Figure 4b**, with linear model fit and confidence bands depicted by blue and yellow lines, respectively. Average intensity values for the homogeneous dye delivery method are depicted by the large, red-filled circle on both graphs, demonstrating the greatly expanded range of combined dye concentrations by the point injection method. These results are also independent of the specifics of injection point location and geometry, as demonstrated in **Figure 4c** and **Figure 4d**, where we compare raw intensity data from two dyes in two different experiments involving two different pairs of injection site locations.

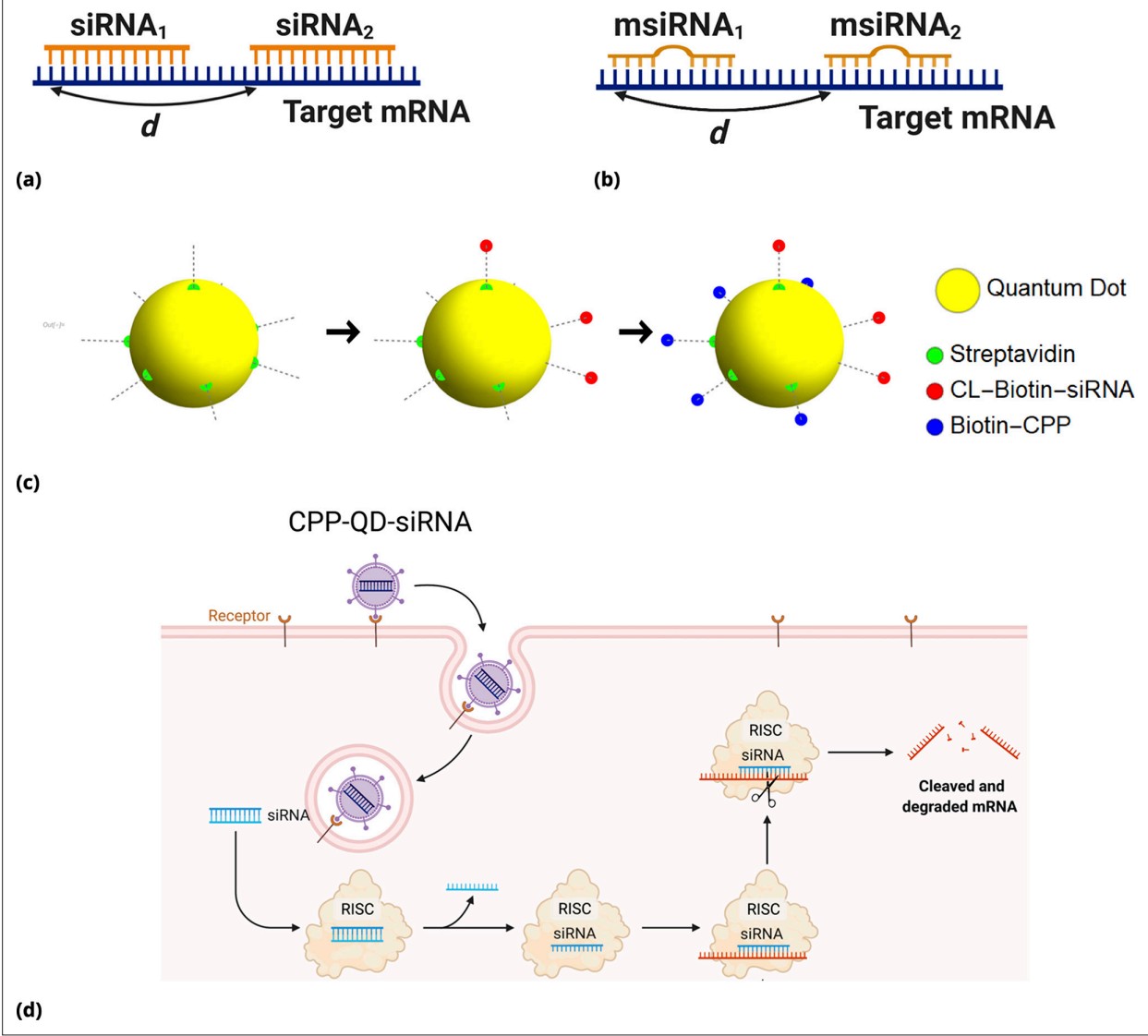

**Figure 6.** siRNA design: Different GFP messenger sites in open frame are targeted by either (**a**) fully complementary siRNA or (**b**) partially complementary msiRNA small interfering RNA molecules of length 21 nucleotides. The distance $d$ in nucleotide units is measured between the 3' ends of targeting sites. The siRNA are non-overlapping if $d \gg 22nt$ and overlapping otherwise. Note that pairs {siRNA$_i$, msiRNA$_i$} completely overlap, $d = 0$. Each siRNA is conjugated to biotin molecules via a cleavable (disulfide) linker (CL). (**c, d**) The siRNA are delivered into cells by Quantum Dot nanoparticles (QD). The QD are functionalized with streptavidin molecules (5–10 per QD nanoparticle) which are used to load siRNA and biotinylated cell penetrating peptides (CPP) guiding and anchoring the fluorescent nanoparticles into the cells. The siRNA sense strand was designed with a cleavable linker (CL) at 5' end. The linker region contains a disulfide bond, which releases the siRNA within the reducing cytosolic environment.

We also observed a high degree of linear correlation in the uptake of co-delivered dyes in fixed cells (*Figure 4e*). As in the case of live cells, homogeneous dye delivery to fixed cells can be predicted well by linear regression obtained from the point co-delivery data (*Figure 4e*). The correlation in uptake ($\rho = 0.79$) for point co-delivery in fixed (suspension) cells is comparable to the correlation observed in experiments with live cells ($\rho \geq 0.8$).

To go beyond the 'toy' system of conjugation-free co-delivery in which we treat one of the diester dyes or quantum dot (QD) nanoparticles as a drug, we co-delivered an authentic drug, auranofin, and a conjugation-free tag (Orange CellTracker dye) to human pulmonary artery endothelial cells (HPAEC). Auranofin is a thioredoxin reductase (TRR) inhibitor that has been shown to increase substantially intracellular $H_2O_2$ levels. We utilized HyPer7, a genetically encoded fluorescent probe for $H_2O_2$ detection, as a reporter for the drug's effect on cell phenotype (*Pak et al., 2020*). The co-delivery of auranofin and conjugation-free dye was performed by point injection. The results of the experiment are shown

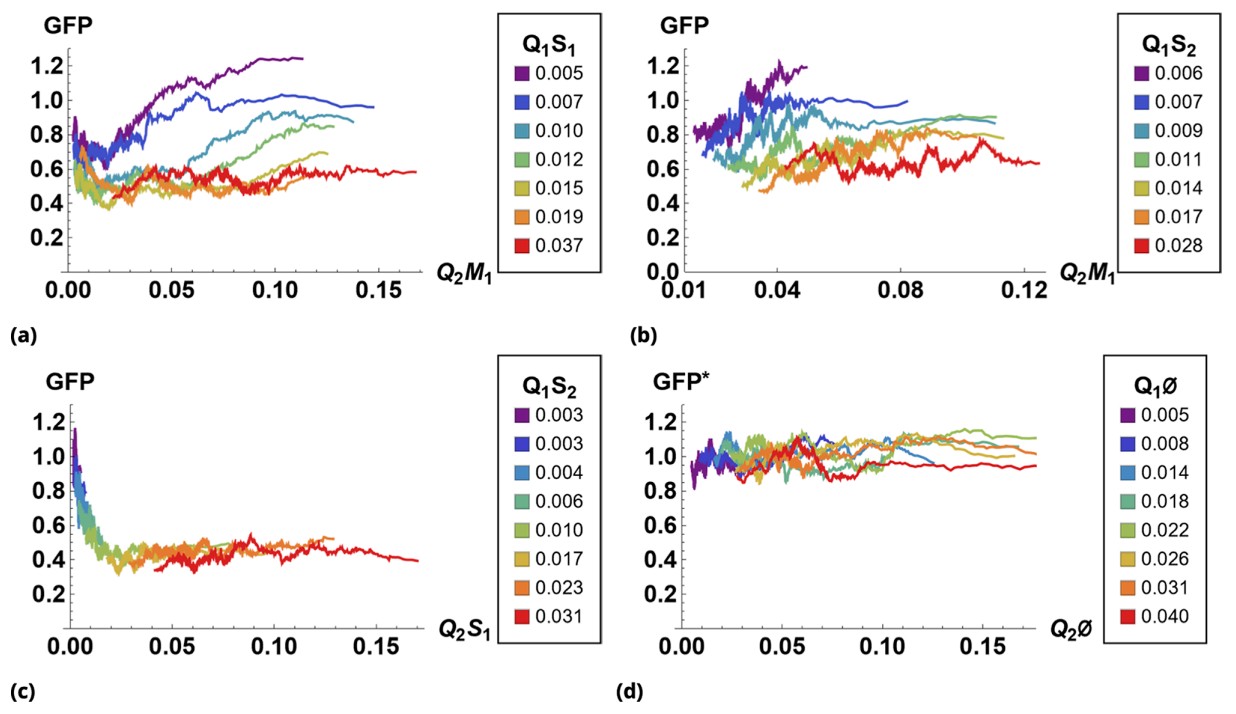

**Figure 7.** The effect of siRNA combinations on target gene (GFP) expression. Three different siRNAs were delivered to cells by optically distinguishable QD nanocarriers ($Q_1$ and $Q_2$) in pair-wise fashion. Two siRNAs, siRNA$_1$ ($S_1$) and siRNA$_2$ ($S_2$), were designed to bind non-overlapping sequences of the mRNA target in completely complementary fashion. The third siRNA, msiRNA$_1$ ($M_1$), has almost identical primary structure as siRNA$_1$ with a single nucleotide mutation close to the seed region. The GFP expression was normalized with respect to control samples. (**a, b**) Relief of GFP repression mediated by msiRNA$_1$ is shown for different concentrations of the repressors siRNA$_1$ and siRNA$_2$, respectively. (**c**) Combination of non-overlapping repressors siRNA$_1$ and siRNA$_2$ results in additive/synergistic repression of target GFP expression. (**d**) Control sample of siRNA free QD combinations was used to account for QD toxicity for both carriers using a machine learning algorithm, and its predicative accuracy is shown here for different concentrations of $Q_1$ and $Q_2$.

in *Figure 5*, where we quantified the fluorescence of the barcode and the $H_2O_2$ sensor's ratiometric reading from the imaging data using a coarse-grained approach.

We observed a significant degree of correlation ($\rho \sim 0.5$) between the barcode tag for the drug and the drug effect as assessed by corresponding fluorescence signals (*Figure 5*). The observed relationship was largely linear with saturation in the ratiometric signal for $H_2O_2$ detection in cells with high auranofin uptake levels.

## Effect of siRNA combinations on gene expression

Our benchmark system (dye combinations) was designed to demonstrate random drug mixing capabilities. We next considered a biological system with a detectable drug mixing effect that could be monitored. To do so, we chose drugs that are direct binding partners of a common, measurable target. We also designed this system to be as portable as possible, that is, be adaptable to different cell types and culture conditions.

We implemented this system as follows: The drugs' target is the exogenous green fluorescent protein (eGFP) gene (d2EGFP, a short-lived variant, cf., Materials and methods). The drugs are small interfering RNA (siRNA) duplexes targeting different regions of the GFP transcript.

We designed two classes of siRNA: (i) typical 21 nucleotide-long siRNA molecules targeting complementary binding sequences on eGFP mRNA; and (ii) 'msiRNA', that is, siRNA with mismatch(es) in its targeting binding sequence. The msiRNA guide strand may still bind target transcripts (and, hence, interfere with target gene expression) via partial sequence complementarity by a mechanism closely mirroring microRNA (miRNA) silencing.

By analogy with endogenous miRNA in eukaryotic cells, one can expect weaker target protein repression by msiRNA compared to siRNA (*Carthew and Sontheimer, 2009*). [Naturally occurring

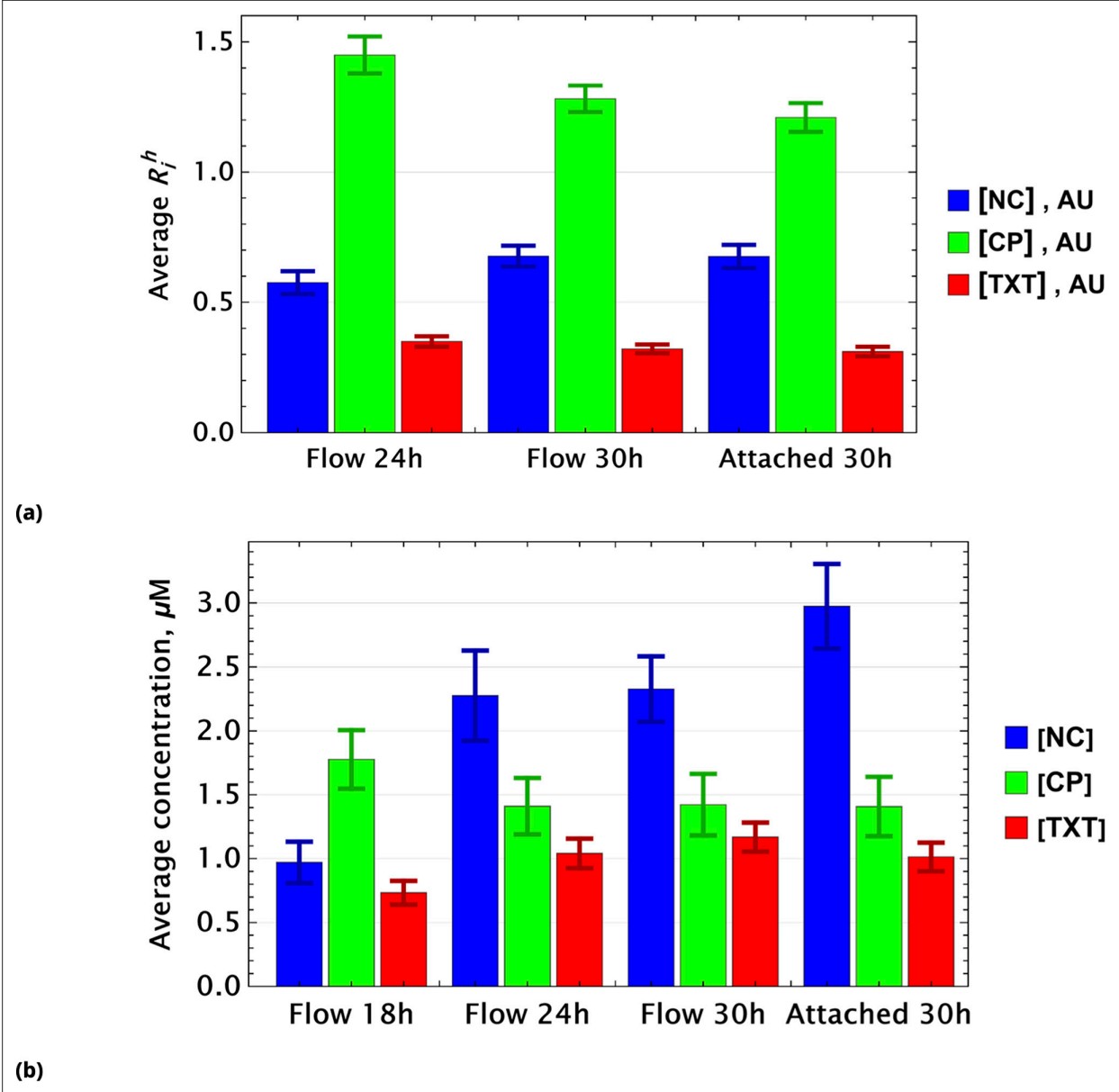

**Figure 8.** Population average of drug concentrations in either detached ("dead") or attached ("live") cells at different time points. (**a**) Population average of $\{[NC], [CP], [TXT]\}$ drug concentrations in homogeneous delivery (thoroughly mixed) experiments, shown in blue, green, and red, respectively. Drugs' concentrations are measured with respect to an auxiliary or marker dye. The first two samples correspond to detached cells harvested from the sample media at different time points and the last sample corresponds to surviving attached cells harvested at the final time point. The error bars represent standard deviations calculated by random partitioning of cell population into bins of 100 cells each. (**b**) The same analysis as in (**a**) applied to non-homogeneous (i.e., gradient or point-delivery methods) drug delivery experiments. Drugs' absolute concentrations were determined using a linear normalization method based on intensities from homogeneously delivered samples. The first three samples correspond to detached cells harvested from the sample media at different time points, and the last sample corresponds to surviving attached cells harvested at the final time point. The error bars represent standard deviations calculated by random partitioning of cell population into 3 bins (average drug concentrations in each bin were considered independent sample measurement).

miRNAs can typically achieve significant repression of the target gene only by utilizing multiple/tandem binding sites on the target mRNA].

We created more diversity in the drugs' action by introducing two different classes of siRNAs targeting different sequence locations: (i) well separated and (ii) overlapping. The motivation behind this design followed theoretical considerations.

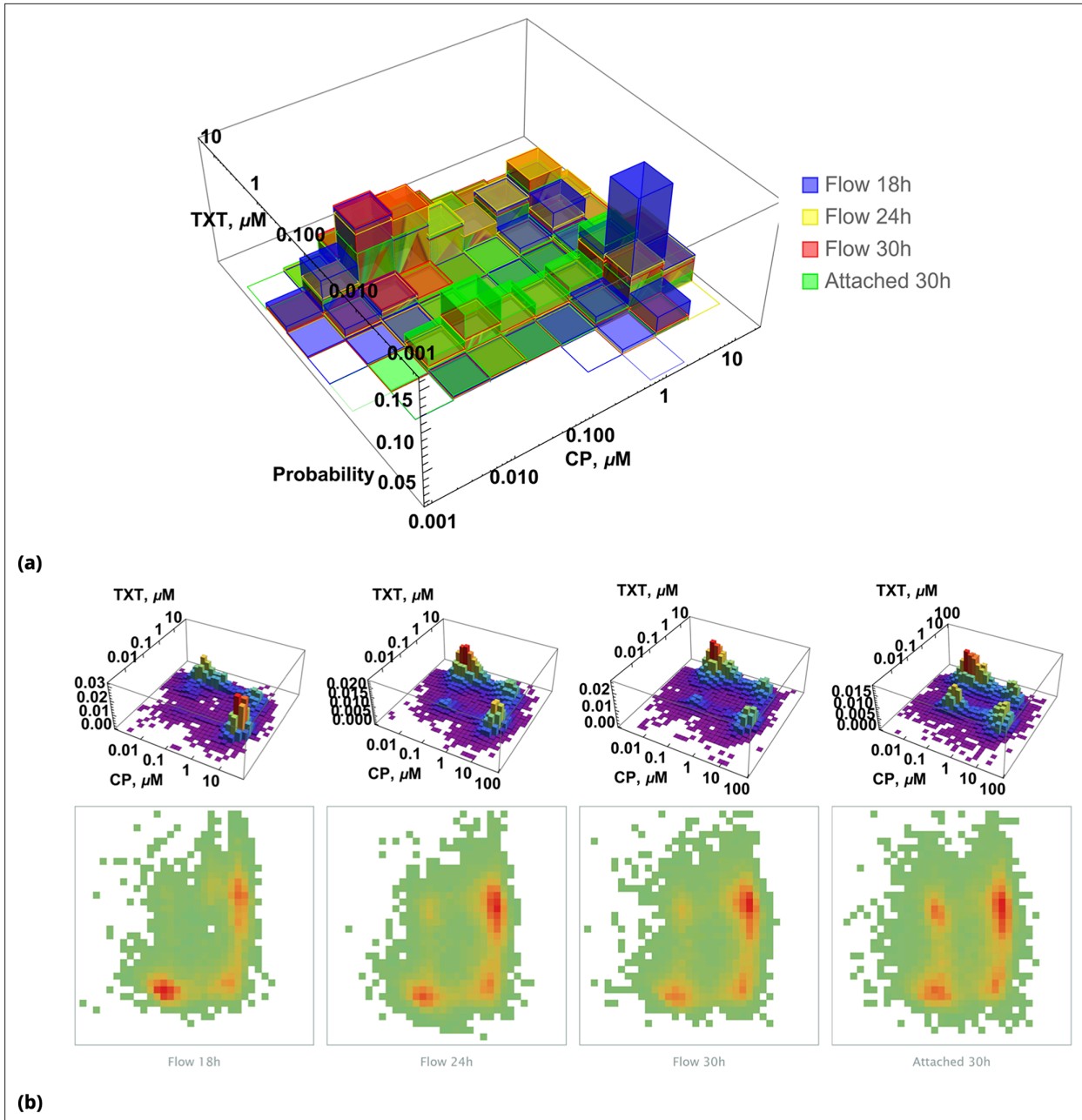

**Figure 9.** Pairwise drug distributions in intact cells in either detached or attached cells at different time points. (**a**) Histogram of $\{CP, TXT\}$ pairwise drug distribution in intact cells. Different colors correspond to detached cells harvested from the sample media at different time points (shown in blue, yellow, and red), and surviving attached cells harvested at the final time point (shown in green). (**b**) Time resolved changes in drug content of dead (first three samples) and live (last sample) cells collected. The density plots represent cell count as a function of drug concentrations (log scale) at different time points.

For low drug concentrations, we expected additive drug interaction regardless of the regions' proximity since there is an abundance of target copy numbers to which siRNA molecules can bind. At high drug concentrations, we expected different drug interaction patterns that depend on target site locations. Different siRNA molecules targeting overlapping regions may, for example, antagonize each other due to binding competition.

For siRNAs targeting non-overlapping regions, one can expect additive or, perhaps, synergistic drug interactions owing to accelerated mRNA degradation and modulated translation rates in the presence of multiple RISC complexes bound to the mRNA. The siRNA design principles, specific

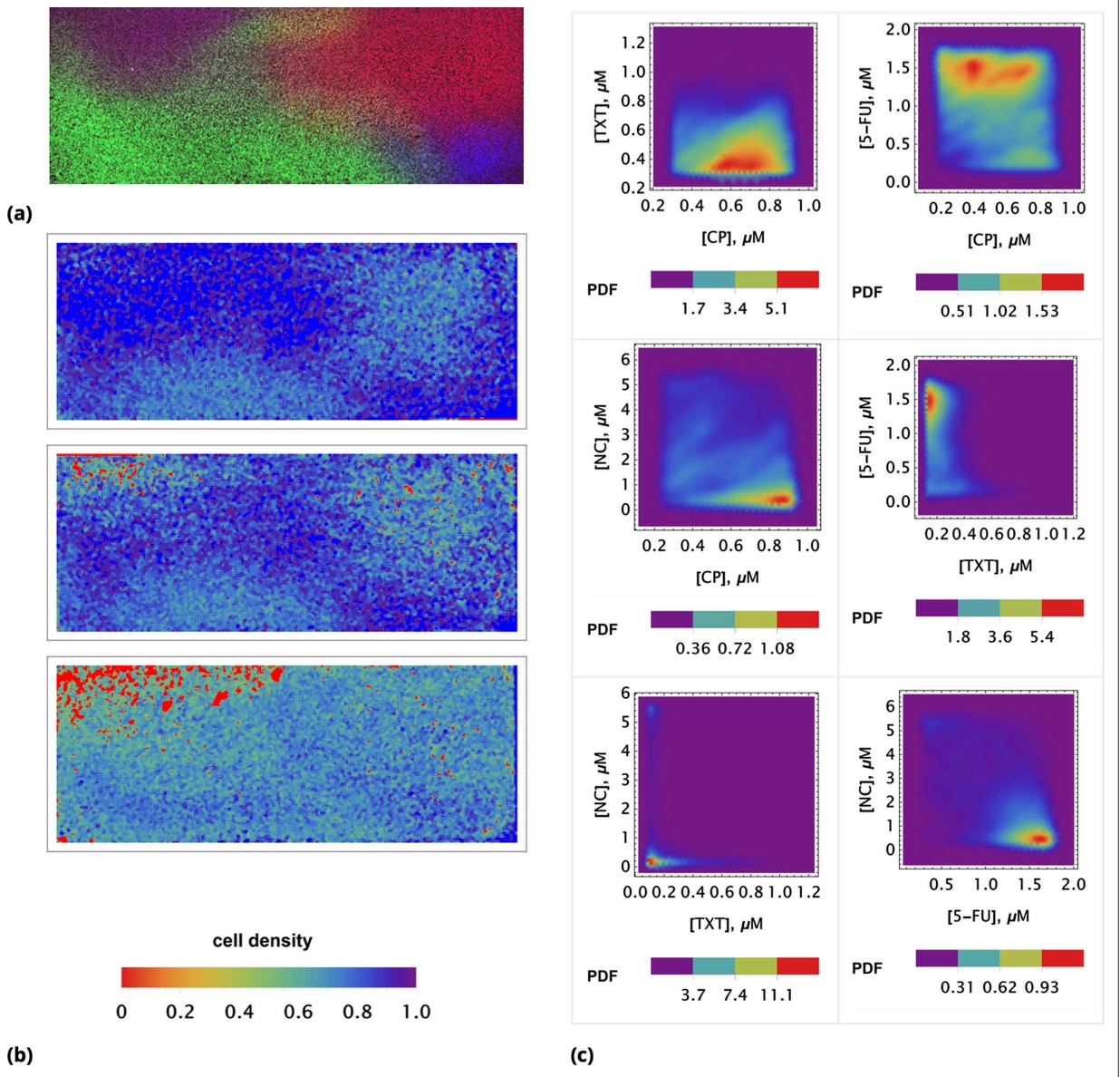

**Figure 10.** Time resolved measurement of attached cell density. (**a**) Initial spatial distribution of drug concentrations of NC, CP, TXT, and 5-FU is shown in blue, green, red, and dark red, respectively. (**b**) Rows 1, 2, and 3: calculated cell local cell density at time points 18, 24, and 30 hours, respectively. Local cell density is determined by the fraction of cell-covered area within a square of $10^4\ \mu^2$. High and low cell confluency corresponds to a fractional range from 1 (blue) and 0 (red), respectively. (**c**) The pairwise initial drug distributions.

sequences, and corresponding target region locations are shown in *Figure 6a and b* (Materials and methods for the details).

Cell-penetrating peptide-conjugated, nanoparticles, Quantum Dots (QD), were used both for drug delivery and barcoding (*Figure 6c and d*). We delivered QD-conjugated siRNAs targeting GFP mRNA in a pairwise fashion. In these experiments, we point-injected all possible combinations of three different siRNAs: siRNA$_1$, siRNA$_2$, and msiRNA$_1$. As expected, point injection of a drug conjugated and delivered in this way introduces a broad range of drug uptake across the sample (*Appendix 1— figure 10*), as compared to the homogeneous random mixing approach (*Appendix 1—figure 9b*), and suppresses the proportionality bias in drug combinations.

The siRNA$_1$ and siRNA$_2$ species have perfect complementarity to non-overlapping regions of mRNA. The msiRNA$_1$ was designed to bind to the same region of mRNA as siRNA$_1$ but has a mismatched nucleotide pair within the targeting sequence. The siRNAs were barcoded using two optically different

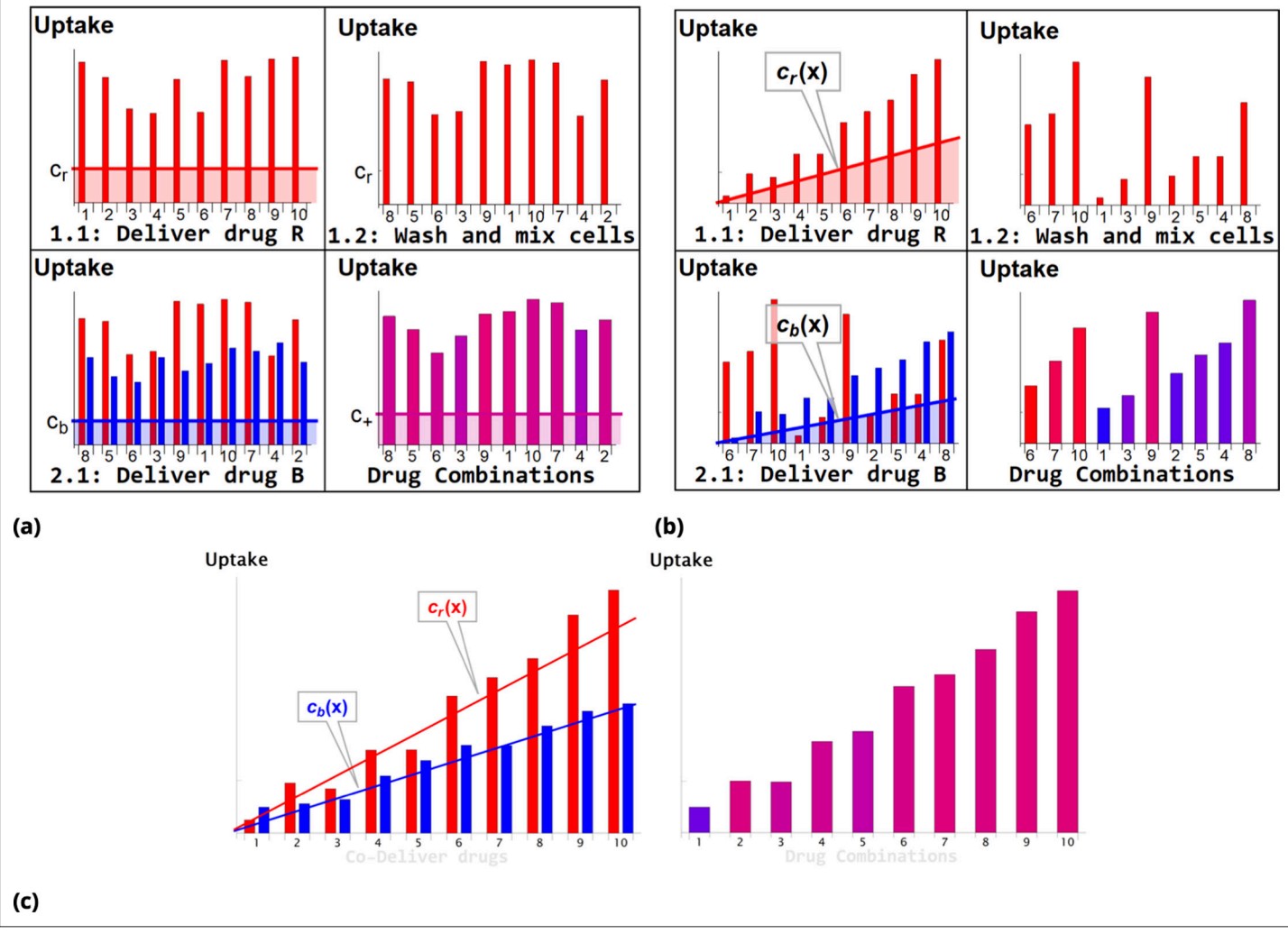

**Figure 11.** Schematic of dye/drug delivery steps for homogeneous and point injection methods (suspension cell culture). (**a**) and (**b**), Graphic idealized illustrations of dye/drug delivery are shown for homogeneous and point injection methods, respectively. Here, x-axis values correspond to the *initial* position index of individual HeLa cells. Media dye concentration profiles are depicted by shaded regions on the left panel for both delivery methods. The y-axis values represent cellular dye uptake for each stage of staining (panels 1.1, 1.2, and 2.1). The final dye combinations are shown in the bottom right panels for both delivery methods (the bar color and the amplitude reflect the molar ratio and total uptake of dyes, respectively). Each wash step is mimicked by reshuffling the cell position index (cf., panels 1.1 and 1.2). Variation in dye uptake due to intrinsic cellular heterogeneity is illustrated for each of the delivery methods by imposing multiplicative noise. For simplicity, the noise in uptake is assumed to be independent for each dye. (**c**) Graphic illustration of the point co-delivery method.

QD nanocarriers, $QD_{605}$ and $QD_{655}$. To demonstrate that the drug combination effect is independent of the tag, we exchanged the tags for each siRNA mixing experiment we performed.

Inherited randomness in cell behavior is not only responsible for variable drug uptake, but also all but guarantees variability in response to drug treatment by individual cells. Even under conditions of identical uptake of a drug and the same exposure time, one expects heterogeneity in individual cell response/phenotype. To overcome this source of heterogeneity, we perform averaging over many cells within each sub-populations characterized by a specific combination of tags (and hence drugs). The number of cells in each subset must be large enough to suppress the variability in drug response ensured by the Law of Large Numbers.

The results of siRNA combinations on target gene expression are shown in *Figure 7*. Here, we partitioned the QD tag intensities into the 2D bins spanning effective parametric space. GFP intensity was averaged across a subpopulation of cells in each bin. Since the QDs at high concentrations lead to cytotoxicity, GFP intensity for each condition was normalized to the control sample in which we

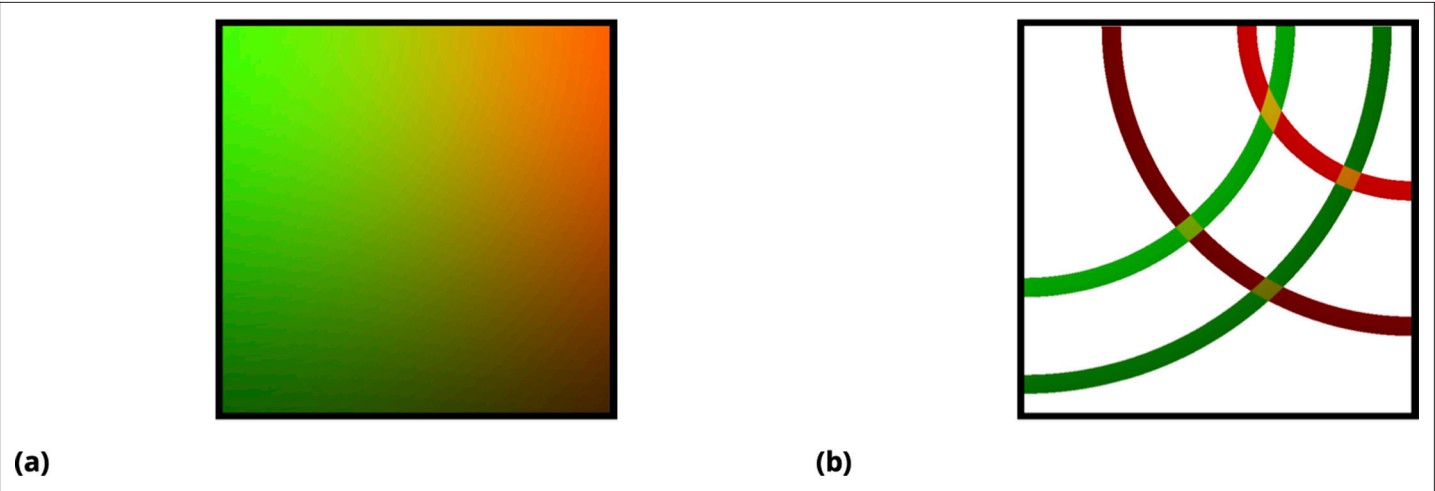

**Figure 12.** Graphic illustration of the point injection method for adherent cells in culture. (**a**) Graphic illustration of the point injection method for adherent cells in culture. The idealized spatial distribution of two drugs (dyes) is shown, assuming they were delivered in the left and right upper corners (green and red color intensities correspond to drug uptake by cells). The fluctuations due to environmental factors and cell heterogeneity are ignored for simplicity, as well as the ripple effect due to finite well size. (**b**) Discretization (binning) of uptake for each drug corresponds to spatial partitioning by concentric annuli of finite width around the point injection origin. The intersection of the annuli for two different drugs corresponds to unique drug combinations (four different conditions, or annulus intersections, are shown).

point-injected $QD_{605}$ and $QD_{655}$ nanoparticles not conjugated to siRNA. (This strategy is equivalent to normalization of repression levels using scrambled siRNA samples as controls.) We used a machine learning algorithm to predict the normalization factor for QD combinations using this control sample (see Supporting Information for details).

At high concentrations, the "mismatched" siRNA, $msiRNA_1$ (with incomplete complementarity in its target binding sequence), acts as an inhibitor for the strong repressors $siRNA_1$ and $siRNA_2$, as shown in *Figure 7a and b*. This antagonistic effect is more pronounced in the $siRNA_1$-$msiRNA_1$ combination as compared to the $siRNA_2$-$msiRNA_1$ combination.

The outcome of pairwise combinations of the non-overlapping strong repressors $siRNA_1$ and $siRNA_2$ suggests additive or synergistic interaction for these drugs (*Figure 7c*). The bias in the titration range of the siRNA concentrations in this experiment is due to limitation of effective mixing conditions (see Supporting Information for details). The predictive accuracy of the machine learning algorithm for GFP normalization (due to toxicity of QDs) is shown in *Figure 7d* for a broad range of mixing conditions.

Here, we, once again, employed the tiled image renormalization approach, which is especially well suited for point injection data analysis because local gradients in drug concentrations persist over scales sized larger than that of a single cell. Since tiles in the image relate to cells in the same spatial proximity, tiled renormalization is equivalent to concentration(s) gating by the local microenvironment.

The results of the pairwise drug combinations in *Figure 7* demonstrate both additive and antagonistic effects of the siRNA combinations. We note that direct binding competition is not the only source of possible antagonistic siRNAs interaction. The repression by siRNA is mediated by the RNA-induced silencing complex (RISC). At the high intracellular concentration of siRNA duplexes, one may expect competition for RISC integration by different siRNA species due to the limited pool of RISC in the cell. This type of competition may result in antagonistic siRNA interactions, regardless of (the abundance of) their target (*Castanotto et al., 2007*; *Loinger et al., 2012*).

## Random mixing and response quantification of three and four drugs

To demonstrate that our approach can be applied to combinations of more than simply pairs of drugs, we considered the following application. We tested drugs commonly used in combination chemotherapy regimens to treat different forms of cancer, such as cyclophosphamide (CP), docetaxel (TXT), 5-fluorouracil (5-FU), and cisplatin (CIS). The desired phenotype in this application is cell death, which can be assessed by utilizing fluorescent biomarkers, such as Annexin-V stain.

Alternatively, death phenotype can be assessed by monitoring cell detachment and/or changes in cell morphology, which may be resolved by bright field/phase contrast microscopy. Cell shrinkage due to dehydration and cell debris can also be detected by a decrease in intensity of the forward light scatter (FSC) signal in flow cytometry measurements (*Wlodkowic et al., 2011*). Unlike annexin-V staining, these cell death detection methods do not require, in principle, additional fluorescence biomarker(s) and, hence, do not reduce the necessary bandwidth required to resolve quantitation of drug combinations in individual cells.

We used the HeLa cell line as a benchmark system to evaluate the effect(s) of drug combinations in individual cells. We utilized either time resolved confocal microscopy, flow cytometry, or a hybrid of both detection methods in our experiments. In all experiments reported below, cells were treated with drugs for a short duration (1/2 hr). Hence, drug concentrations were chosen to be significantly higher than their corresponding typical $IC_{50}$ values. (In typical experiments, drug(s) incubation is 48 hr, roughly 100 times longer than in our setting.) In order to create a negative drug control, cisplatin was inactivated by DMSO (*Hall et al., 2014*).

We note that our goal was to demonstrate the technical feasibility of our stochastic mixing approach. A more practical, generic application (in terms of both an expanded list of drugs and of cell lines) and biological data interpretation are beyond the scope of this sudy.

## Flow cytometry measurements of cell death phenotype, three-drug combinations

In order to assess drug combinations leading to cell death using flow cytometry, we harvested detached cells and debris in the sample media at different time points and analyzed the fluorescence intensities of the specific drug tags, which correspond to *effective* drug combinations. At the final time point of the experiment, we harvested the remaining attached cells and analyzed the fluorescence intensities of the specific drug tags similarly, which correspond to *ineffective* drug combinations.

We stained cells homogeneously with the same auxiliary dye (Deep Red CellTracker dye) not conjugated to any drug. We used an identical concentration of this dye in all samples, including controls. The CIS, CP, and TXT drugs were labeled with blue, green, and orange CellTracker dyes, respectively. All CellTracker fluorescence probes (except for Deep Red) contain a chloromethyl or bromomethyl group that reacts with thiol groups, utilizing a glutathione-S-transferase–mediated reaction.

In most cells, glutathione levels are high (*mM* range) and glutathione transferase is ubiquitous. The CellTracker Deep Red probe contains a succinimidyl ester functionality, which reacts with amine groups present on proteins, and, hence, is also ubiquitous in cellular cytoplasm. The ratio $R_i$ of any CellTracker dye *i* signal to Deep Red intensity can be viewed as a proxy to cellular concentration of the CellTracker and its cognate drug, since the Deep Red signal is proportional to cell volume.

We assumed that for sufficiently large cell fragments, this normalization also holds, which allows an estimation of the drug concentrations in both intact cells and dead cells (debris). We note that due to stochastic uptake of any dye including an auxiliary dye, this normalization is only meaningful after averaging over a significant number of detection events required to suppress the signal noise.

In order to convert the ratio $R_i$ to absolute drug concentration, we calculated $R_i^h$ for a homogeneously delivered (with mixing) drug sample where the drug concentrations are known exactly. The final concentrations of drugs in these homogeneously delivered controls were {1, 1, 0.1 $\mu M$} for NC, CP, and TXT, respectively.

The factor $R_i/R_i^h$ calculated for individual cell is a proxy for the volume concentration of the drug *i* compared to homogeneously delivered drug control. Given the ubiquitous nature of all CellTracker dye targets, one expects that this estimate will hold even after cell division. The absolute fluorescent signal from the parent cell will decrease due to dilution by cell division, but the ratio of homogeneously distributed targets should persist.

To demonstrate that this expectation holds, we considered dynamic changes in fluorescence signal intensity in the controls of homogeneously delivered drugs. The population average factors, $R_i$ for i={NC, CP, TXT}, are shown in *Figure 8a* at different time points.

Small time-dependent modulation of $R_i^h$ values are noted and are a consequence of the stochastic nature of drug or dye uptake, a process that ensures the existence of cells with a distribution of drug concentrations, including importantly some that take up less drug(s) than the cell average. The corresponding $R_i$ values for these cells is smaller compared to cells that received higher drug dosage.

Hence, this low $R_i$ subpopulation is more likely to survive longer. Since this dynamic drift is rather small, however, we neglect it in further analysis.

For samples created by gradient drug mixing, one naively expects qualitatively the following picture for the population average of drug concentrations: the concentrations of effective and ineffective drugs should decrease and increase, respectively, with time. This expectation derives from the fact that the subpopulation of cells that received ineffective drug concentrations or drug combinations becomes increasingly dominant over time.

This qualitative picture (evaluated on a whole cell population level) holds only if drugs act independently in individual cells. Drug pharmacodynamics may also be non-uniform over time. We observed these behaviors in the gradient-derived mixed sample data shown in *Figure 8b*.

The dynamics of inactivated cisplatin (labeled NC in *Figure 8b*), indeed, displays the time dependence expected from an ineffective drug (concentration). It is much more difficult to interpret the pharmacodynamics of CP and TXT because the time dependence is not monotonic. Drug-drug interactions, drug effect on cell cycle, and other time-dependent drug actions can all play a significant role in overall cell population behavior.

To visualize the interplay of drugs on individual cells, we first reduced data dimensionality by discarding information about the level of [NC] in cells (assuming DMSO inactivated cisplatin completely). Under these circumstances, one can plot the probability distribution, $P(x, y)$, to find cells with different drug combinations $\{x = [CP], y = [TXT]\}$ at different time points, as shown in *Figure 9a*. Two dimensional projections of this histogram for each individual time point are shown in *Figure 9b*.

Our overall goal was to demonstrate the technical capability of the stochastic mixing approach. More practical application (in terms of both list of drugs and cell lines) and data interpretation are beyond the scope of this study.

## Microscopy measurements of cell death phenotype, four-drug combinations

For microscopy-based measurements and image analysis, we used the point injection method to create local drug gradients across the sample of confluent HeLa cells on a glass slide. Followed by the initial imaging of the drugs' distributions across the sample, we monitored cell detachment (and changes in morphology, e.g. shrinkage) at different locations on the slide at different time points. The estimated intracellular concentrations of all chemotherapy drugs used were assessed using the non-conjugated fluorescent tag (dye) method described above.

The image analysis allows one to correlate the initial drug combination concentrations in each locale on the slide and subsequent death phenotype in the same position at different time points. In principle, cell migration between locales can complicate the analysis, but we used confluent seeding conditions to minimize this effect.

The outcome of a typical microscopy experiment is shown in *Figure 10*. Here, the initial distribution of each drug across the sample is shown in *Figure 10a* using four different colors (blue, green, red, deep red) corresponding to the fluorescent label for each drug. Local cell density at different time points is shown in *Figure 10b*, using an inverse rainbow color scheme (where blue and red correspond to fully confluent cells and no cells, respectively). All possible pairwise initial drug distributions are shown in *Figure 10c*, see also *Appendix 1—figure 16* in Supporting Information.

We used a coarse-grain approach for image analysis, averaging fluorescence intensities over an area of $\sim 10^4 \, \mu m^2$ at each locale. Cell density was assessed by cell segmentation using bright field imaging data and the calculation of the area fraction covered by cells in each coarse-grain locale. The absolute drug concentration can be determined using the homogeneous delivery method discussed above.

## Discussion

Biologists often treat noise in samples as a nuisance that must be minimized, an important consideration for typical experimental measurements. Here, we rely on engineered local heterogeneity in drug uptake as a tool to encode drug treatment regimens and to predict the macroscopic response to drug perturbations. This approach was inspired by the celebrated fluctuation–dissipation theorem

in statistical physics (*Cardy, 1996*) that predicts the response of a system to external perturbations based on the underlying properties of noise in the system at equilibrium.

Rather than testing thousands of samples, as little as a single-cell culture sample is required to perform the search for optimal drug combination using our approach. Applications of our method are not limited to the identification of optimal drug concentrations, as the approach is applicable to any quantifiable cell phenotype. The method is readily generalizable to a much broader set of problems involving complex biochemical or molecular perturbations applied in vitro and in vivo. Examples of these perturbations include combinations of signaling molecules, optimization of drug delivery strategies, lineage differentiation identification, stem cell manipulations and tracking, etc.

The best possible random drug mixing is achieved in cells in suspension. Suspension cells can be easily mixed (i.e. cell locations can be randomly re-distributed), and local drug gradients can be applied in a statistically independent serial fashion. By contrast, since the locations of adherent cells in the sample are fixed, we exclusively rely on the randomness of the drug distribution in the media by means of forced advection, which is challenging to control. Hence, further theoretical and experimental work is required to improve the stochastic drug mixing range and its smoothness for adherent cells. A possible way to improve the mixing range (and minimize reagent use even further) can be utilization of recently developed 'Dye Drop' method (*Mills et al., 2022*).

We would like to point out an important limitation of our stochastic drug mixing method (at least in its current state). Throughout this paper, we tacitly assumed that both drugs and their labels are retained by the cells and are not re-distributed from cell to cell after the washing step. This is certainly the case for siRNA drugs delivered via nanoparticles and CellTracker dyes in our applications, but this assumption does not necessarily hold in general.

Drugs can, in principle, leak from the cells into the culture media after the washing step and, eventually, be redistributed homogeneously across the entire cell population. This behavior will affect the 'area under the curve' for individual cells after prolonged incubation periods, especially cells that had initially very low (or no) drug exposure. We assumed that this effect is small and did not study the kinetics or the extent of this 'secondary' drug exposure here. (Multiple washing steps were performed to mitigate this effect.) More careful analysis and 'compensation' is required for drugs that are not retained by the cells.

Another possible drawback of our method is a limit on maximal drug concentration due to drug solubility and its solvent (e.g. DMSO) toxicity. Since we utilize short drug exposure times, high drug concentrations may be required to test a broad range of conditions. Typically, low concentrations of individual drugs are desired in combination therapies and, hence, our method can be implemented even for drugs with low solubility.

We expect possible adaptation of the point injection method to facilitate random mixing in vivo. The spatial distances between injection sites, geometry, needle gauge size, injection volume, and the specific measurement protocol (readout) will, of course, all need to be established and optimized for a particular application.

Throughout this paper, we focus exclusively on the measurement approach rather than on data interpretation, which is context- and phenotype-dependent. Further work is necessary to develop a methodology, interpretation, and dedicated numerical methods to identify optimal drug combinations from high dimensional experimental data.

## Materials and methods
### Drug labeling and delivery

The workflow of the temporal local gradient method used in this work is depicted schematically in the *Figure 1*. The cell culture sample is placed on a horizontal vibration-free surface, and buffer containing labeled drug is injected in the periphery of the sample. The ratio of injection to sample volumes was typically $1-5\%$ and injection speed is slow to ensure local initial distribution of the drug in the sample. (In order to visualize and adjust injection volume, release speed, and dispersion dynamics, dyes visible by naked eye or food-dyes such as Brilliant Blue FCF can be used on training samples.) Care should be taken not to disturb the sample during the incubation period (to avoid sample homogenization). After the incubation period, culture media should be quickly replaced with fresh media in adherent cell culture samples.

In suspension cell experiments, cells are placed into narrow cuvettes using a buffer density lower than cell density. After cell sedimentation, their spatial distribution corresponds to a narrow rectangle on the bottom of the cuvette. Local point injection is delivered in the 'corner' of this rectangle sufficiently slowly so as to minimize the cells' movement in the cuvette. Mixing labeled drugs with iodixanol (OptiPrep), an inert liquid with a density higher than PBS/media, can be used to minimize dispersion of local injection in the vertical direction (*Mills et al., 2022*). Immediately after the drug incubation, large wash volume is used to minimize re-distribution of drug(s) within the sample prior to and during spin down step.

We used multiple different dyes to perform the multiplexing experiments: CellTracker dyes (blue, green, orange, CMTPX red, and deep red) and CellTracer dyes (violet, carboxy fluorescein succinimidyl ester [CFSE], yellow, red, and far-red). The CellTracker probes have been designed to pass freely through cell membranes, and once inside the cell, they are converted into impermeable fluorescent products. The probes are transferred to daughter cells, but are not transferred to adjacent cells in a population. The CellTracker fluorescence probes contain chloromethyl/bromomethyl or succinimidyl ester reactive groups that react with ubiquitous intracellular thiol or amine groups, respectively.

More flexible barcoding can be achieved by utilizing a tag bound to a drug by a cleavable linker. The tag can be designed to be retained by the cell, with the delivered cellular drug amount proportional to the tag's fluorescence signal. We used nanoparticle drug delivery to apply this labeling method, namely, cell-penetrating peptide-conjugated quantum dots (QD) were used both for drug delivery and barcoding. The fluorescent reporter (QD) is retained within the cell upon entry and the drug, siRNA in our case, is released locally into the cytosol via cleavage of a disulfide bond in the cytosolic reducing environment.

Finally, we utilized the point injection method to co-deliver locally both drug and conjugation-free fluorescent dye retained by the cells. We note that to achieve one-to-one (monotonic) correspondence between total cellular drug uptake and its conjugation-free tag, it is important to ensure non-saturation labeling and detection conditions. If the tag fluorescence intensity saturates beyond a certain exposure concentration (or time), the total drug uptake prediction will be inaccurate. Most of the dyes we used (CellTrace and CellTracker) have abundant intracellular target concentrations and saturate only at very high extra-cellular concentrations ($> 1\,mM$). Therefore, the saturation in tag signal may be caused by microscopy detection limits (for either too low or too high fluorescent signals at the given laser power and sensitivity of the photomultiplier settings). Careful adjustment of these parameters is required to ensure a maximal dynamic detection range.

To avoid label and drug homogenization during the wash/collection steps using media (containing both fluorescent tags and drugs), the washing steps need be performed quickly using large dilution volumes. This homogenization, in turn, can affect local labeled drug gradients in the culture, reducing the dynamic range of drug concentrations within the sample. Cells that were initially exposed to very low local drug(s) concentration are most affected. We note that this short time scale 'leakage' effect occurs with both label(s) and drug(s) proportionally. Hence, label intensity should be proportional to the corresponding drug concentration, and can be still used as a proxy for drug concentration.

Long time scale leakage occurs if the drug within cells is released to neighboring cells either via media or through cell-to-cell contact. In general, this secondary drug exposure cannot be accounted for using fluorescent tags in our method because the tags/labels are designed to be irreversibly bound to the cells and do not leak (labels can, however, be diluted by cell division). Drug leakage from cell to cell mediated by media can be estimated using the volume ratio of cells and culture media. This ratio in most cell culture condition is negligibly small and leakage of this sort can be ignored.

Drug leakage from cell-to-cell mediated by direct contact can be much more significant. However, this effect is mitigated in adherent cell culture experiments by the coarse-graining procedure when the optical field containing multiple neighboring cells is averaged.

## Effective drug combinations by stochastic mixing

Steps involved in the point injection method compared to homogeneous delivery are shown in *Figure 11a* and *Figure 11b*, respectively. Cells in suspension culture were placed in a cuvette and a dye $A$ (Red Dye in the example shown in *Figure 11b*) was injected locally (just beneath the meniscus). The volume of the point injection must be small compared to the cell culture volume in the cuvette

(we used $\sim 1:20$ volume ratio). (N.B., the graphic illustration in *Figure 11b* is an idealization because the gradient created by point delivery is not spatially linear.)

The cultured cells suspended in the cuvette were left undisturbed during the short incubation process (10–30 min). The transient gradient of dye $A$ results in higher dye uptake by cells located closer to the point injection site. The subsequent wash step ensures removal of remaining dye $A$ and also mixing of differentially stained cells. The process is then repeated with another dye $B$ (Blue Dye in the example shown in *Figure 11b*) in exactly the same fashion as with dye $A$.

As a result, a broad range of drug combinations can be achieved. Consider, for example, cells in the proximity of the dye $B$ point injection site. These cells exhibit a wide range of dye $A$ uptake and high uptake of dye $B$. By contrast, cells remote from the dye $B$ point injection site are also characterized by a broad range of dye $A$ uptake but low uptake of dye $B$. The conjugation-free barcoding method to co-deliver locally both drug and fluorescent dye retained by the cells is illustrated graphically in *Figure 11c*.

The local injection method was used to create stochastic drug combinations in adherent cell culture as well. (Cell mixing, naturally occurring in suspension cell culture, is not feasible in this case unless cells are detached and re-seeded, actions that require time and may interfere with drug response.) The local drug gradients across the single sample created using this method are driven mostly by initial drug dispersion (forced advection) during the injection if the incubation time is short (cf., Supporting Information for technical details on diffusion rate of small molecules).

The results shown in *Figures 3 and 4* correspond to experiments performed with 90% confluent HeLa cells cultured in four wells with a chambered coverslip Ph+ (Ibidi). Point injections were performed pairwise: pair (i) CellTrace Violet and Qtracker 655 Cell Labeling Kit (referred to as Blue and QD655 dyes in the main text); pair (ii) CellTrace CFSE and CellTracker Red CMTPX (referred to as Green and Red dyes in the main text).

The total culture volume of each chamber was 750 $\mu l$ and 30 $\mu l$ were replaced with a pre-mixed pair of dyes injected at one of the chamber corners. The process was repeated with the second pair of pre-mixed dyes injected into another chamber corner. The manual injection rate is difficult to control but the total release time was $\sim 1$ s. The incubation was performed at room temperature for $\sim 20$ min in the cell culture hood (with all possible precautions taken to avoid shaking the slide).

The graphic illustration in *Figure 12* demonstrates this approach for a pair of drugs. Here, an idealized drug mixture distribution is depicted by color and intensity in *Figure 12a*. All fluctuations due to cell variability, advection, and non-independence in drug uptake are ignored in these illustrations for simplicity. Moreover, we assumed that transient drug uptake gradients are symmetric (radial) with respect to the point injection locations (left and right top corners of the rectangular slide). In reality, of course, all of these assumptions, including radial symmetry, are violated, resulting in much more random behavior, which is precisely what we wish to exploit to enhance the heterogeneity in drug uptake across the cell population.

In *Figure 12b*, the origin of the distribution as a linear superposition of individual drug uptakes is shown, which is designed to illustrate that unique mixing conditions, even in this idealized case, depend on the geometry of the matrix and point injection locations. Discretization (binning) of uptake for each drug has a simple geometric interpretation in this idealized case, shown as regions of annular intersections in which colors are uniquely mixed. For a real system, this symmetry is violated due to processes such as advection, and we, therefore, must rely on the fluorescence intensities of the tags rather than geometry to partition the ensemble into bins; however, as in the case of the idealized system in *Figure 12b*, one expects a finite-sized local region on the slide where drug concentrations/uptakes can be assumed to be constant.

## Statistical processing of single-cell data

In all single-cell experiments, multidimensional intensity data were compiled using averaged fluorescence intensity measurements of each optical drug barcode for each cell. Additionally, the ensuing phenotype for each individual cell was recorded using the corresponding fluorescence signal in functional experiments (conjugation-free barcoding, auranofin, and siRNA combinations).

For experiments utilizing a single barcoded drug, we used coarse-grained modeling to recover the relationship between drug and tag as follows: all measured single-cell fluorescence intensity data were ordered by increasing value of the intensity of the tag. The ordered data were grouped into

bins (subpopulations) of equal length $r$ (number of cells) large enough to suppress intrinsic noise. After averaging the intensity data in each bin, the dimension of the resulting data array is reduced by a factor $r$. This coarse-grained process ensures reduction in intrinsic noise due to the Law of Large Numbers (cf., Supporting Information). Alternatively, noise filtering can be implemented by moving the average approach, preserving data dimensionality.

A similar strategy was employed for coarse-grained analysis of barcoded multi-drug experiments. Cell intensity data were clustered hierarchically based on the barcodes' signals, and renormalized data were produced by averaging all optical parameters within each cluster (cf., Supporting Information).

Specifically, for siRNA experiments, after image segmentation, fluorescence of QD drug carriers and of GFP was assessed for individual cells. Each drug's uptake was assumed to be proportional to the intensity of the corresponding QD carrier in each cell. The data were typically recorded as a 3-tuple $\{QD_1, QD_2, GFP\}_i$ for each cell $i$.

We grouped cells into bins characterized by different QD combinations (in terms of fluorescence intensities $\{QD_1, QD_2\}$). For pairwise drug combination experiments, binning was achieved by partitioning the 2d space of the corresponding QD-tags' fluorescence values into a finite mesh. The bins were designed to contain 30 or more cells each. We averaged a phenotypic marker (e.g. GFP intensity) over each binned cell population.

## Cell culture and microscopy imaging

The great majority of the experiments were conducted with the HeLa cell line. We used human pulmonary artery endothelial cells (HPAEC) for quantification of auranofin delivery. All experiments utilizing the point injection method were performed in confluent cell culture. The cells were grown on culture plates (microscopy slides) as follows: we used rectangular 4 and 8 chamber slides (Ibidi μ-Slides) for siRNA experiments, and either rectangular or circular slides for dye labeling experiments.

In experiments requiring single-cell imaging, cells were either 'shrunken' or completely detached using Accutase treatment to facilitate image segmentation. Homogeneous dye and QD labeling were performed according to the manufacturer's (Thermo-Fisher) suggested protocol(s), typically at room temperature.

For point injection experiments, we used 10 x dye concentration (compared to the homogeneous staining condition). Delivery was carried out by injection of $\sim 0.05 - 0.1$ cell culture volume locally. Cell imaging was performed with an inverted fully motorized Zeiss LSM-800 confocal scanning microscope utilizing 405, 488, 561, and 640 nm diode lasers for excitation of dyes and markers. Most of the microscopy experiments were performed using the 5 x objective to capture larger optical fields necessary for statistical analysis.

Imaging data from single-cell experiments were processed as follows: cell segmentation was performed using bright-field and/or dedicated cytosolic markers. We employed Cellpose (*Stringer et al., 2021*) for accurate cellular segmentation. For the auranofin co-delivery experiments, the ratiometric signal of the HyPer7 sensor was calculated as the ratio of cellular fluorescence intensities of the GFP channel excited by 488 and 405 nm laser sources.

Excitation and emission settings were chosen according to the dye manufacturer's suggestions. In all QD experiments, we used a 405 nm laser for excitation with corresponding emission spectra filters. We tested fluorescent labels/markers individually to detect possible cross-channel leakage/overlap. This analysis was especially critical in dye co-delivery experiments where the degree of correlation may be strongly affected by channel leakage. To ensure channel 'isolation', we utilized dyes/fluorophores with well separated emission spectra for these experiments.

HeLa cells were maintained in Dulbecco's modified Eagle's medium (DMEM) containing 0.11 g/liter sodium pyruvate, 2 mM $L$-glutamine, 4.5 g/liter glucose, and 10% fetal bovine serum (FBS). Human pulmonary artery endothelial cells (HPAEC) were cultured at 37°C in modified EGM2 medium (Clonetics). Cells were maintained in continuous culture in an air-5% $CO_2$ atmosphere at constant humidity and used within three to five passages. For fixation experiments, after aspiration of the culture media, cells were incubated with 10% formalin for 5 min at room temperature followed by a PBS wash. For cell suspension experiments, we detached HeLa cells with Accutase solution (Sigma-Aldrich), washed the cells with culture media, and kept them at either room temperature or 4°C in dPBS in non-treated plastic cuvettes for a short duration of dye(s) labeling (typically 1/2 hour).

In siRNA experiments, point-injection of two different QD nanoparticles carrying different siRNAs was performed at two different corners of the cell culture plate (chamber slide). Samples were incubated at 37° for 30 min. (Of note, no special precautions were taken to suppress the advection process, which facilitated the creation of concentration gradients).

## siRNA design and delivery method

We used short-lived green fluorescent protein (d2EGFP) to ensure correlation between fluorescent GFP signal and mRNA target concentration in cells. The d2EGFP plasmid (*Warren et al., 2010*) was a gift from Derrick Rossi, Addgene #26821. We used siRNAs that were designed and experimentally validated by Dharmacon (cf., Accell eGFP Control siRNA) and the Sharp laboratory with binding strand sequences GCCACAACGTCTATATCAT ($siRNA_1$) and GCACCATCTTCTTCAAGGA ($siRNA_2$), respectively. The binding sites of $siRNA_1$ and $siRNA_2$ are separated by distance $d = 153$ nucleotides (in primary sequence). The msiRNA1 construct has a single nucleotide mismatch in the binding strand, GCCACAACGGCTATATCAT, and the separation distances to $siRNA_1$ and $siRNA_2$ were $d = 0$ and $d = 153$ nucleotides, respectively.

We utilized a fluorescent tag for each siRNA that was retained by the cell as follows. All siRNAs were designed with a cleavable disulfide linker (CL) attached to their 3' end (*Figure 6*). We used streptavidin-conjugated Quantum Dots (QDs) as fluorescent tags for each class of siRNAs, creating uniquely labeled siRNA-QD pairs. Quantum Dots (QDs) are semiconductor nanoparticles, typically excitable by UV light, with fluorescence emission spectra that are fairly narrow. The emission peaks of QDs depend on the nanoparticle size, a property that makes QDs very useful for high-throughput microscopy measurements (*Wegner and Hildebrandt, 2015*). To facilitate siRNA-QD cellular delivery, we further functionalized the QDs with a cell penetrating peptide (CPP) [a nanopeptide $(Arg)_9$] that can facilitate cell uptake through endocytosis or non-endocytotic mechanisms (*Verma and Stellacci, 2010*) (cf., Supporting Information for technical details).

## Coarse-grained image analysis

In addition to course graining of single-cell data, we develop a similar approach with respect to raw images. We performed effective local averaging over multiple confluent cells by partitioning the entire image into tiles. The size of the tile (typically, ~50x50 microns) was estimated to contain at least a few cells on average.

Each tile in the image was replaced with an effective 'pixel' characterized by the average fluorescence intensity of real pixels within the tile. The only segmentation we performed in this case was identification of large lacunae (cell-free regions) in the optical field, with exclusion of these regions from image analysis.

## Acknowledgements

The authors wish to thank Arvind K Pandey for assistance with the HyPer7 probe, William M Old-ham for numerous insightful suggestions on experimental design, and Stephanie C Tribuna for assistance in preparation of this manuscript.

## Additional information

### Funding

| Funder | Grant reference number | Author |
| --- | --- | --- |
| National Institutes of Health | HGHG007690 | Joseph Loscalzo |
| National Institutes of Health | HL108630 | Joseph Loscalzo |
| National Institutes of Health | HL155107 | Joseph Loscalzo |

| Funder | Grant reference number | Author |
| --- | --- | --- |
| National Institutes of Health | HL155096 | Joseph Loscalzo |
| National Institutes of Health | HL119145 | Joseph Loscalzo |
| American Heart Association | D700382 and CV-19 | Joseph Loscalzo |
| American Heart Association | AHA957729 | Joseph Loscalzo |

The funders had no role in study design, data collection and interpretation, or the decision to submit the work for publication.

## Author contributions

Vlad Elgart, Conceptualization, Data curation, Software, Formal analysis, Investigation, Visualization, Methodology, Writing - original draft, Writing – review and editing; Joseph Loscalzo, Conceptualization, Resources, Supervision, Funding acquisition, Validation, Methodology, Project administration, Writing – review and editing

## Author ORCIDs

Vlad Elgart (ID) http://orcid.org/0000-0002-6102-6904
Joseph Loscalzo (ID) http://orcid.org/0000-0002-1153-8047

## Decision letter and Author response

Decision letter https://doi.org/10.7554/eLife.85439.sa1
Author response https://doi.org/10.7554/eLife.85439.sa2

# Additional files

## Supplementary files

• MDAR checklist

## Data availability

Imaging, flow cytometry data, and custom Wolfram Mathematica computer code use for data analysis were deposited in Dryad database.

The following dataset was generated:

| Author(s) | Year | Dataset title | Dataset URL | Database and Identifier |
| --- | --- | --- | --- | --- |
| Loscalzo J, Elgart V | 2022 | Local Generation and Efficient Evaluation of Numerous Drug Combinations in a Single Sample | https://dx.doi.org/10.5061/dryad.76hdr7t1d | Dryad Digital Repository, 10.5061/dryad.76hdr7t1d |

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

# Appendix 1

## Combinatorics of drug combinations

Test this *Appendix 1—figure 1a* and *Appendix 1—figure 1c*.

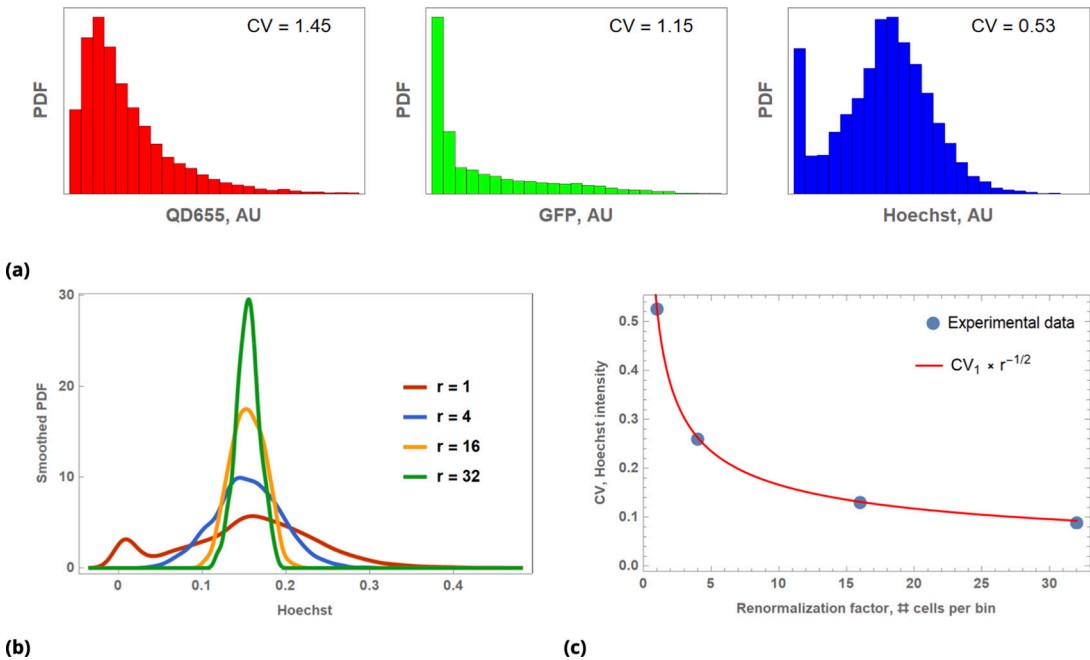

**Appendix 1—figure 1.** Illustration of Law of Large Numbers applied to single-cell marker fluorescence intensities: (**a**) Large degree of variability for different cellular phenotypes is shown for QD, GFP, and Hoechst dye single-cell fluorescence intensity data. Each histogram represents a PDF distribution of mean pixel intensity for an ensemble of individual cells. (**b**) Smoothed distribution of Hoechst fluorescence intensity for different levels of renormalization. The red curve corresponds to single cell measurement statistics, $r = 1$, of raw data. (**c**) Coefficient of variation, $CV$, of a statistical ensemble is plotted as a function of scaling factor $r$. Here, we defined $CV_1$ as a coefficient of variation in Hoechst fluorescence intensity for *raw* single cell data ($r = 1$).

The number of drug combinations, $N_c$, for a given number of drug doses can be expressed as

$$N_c = \prod_{i=1}^{i=N_d} (n_i + 1),\qquad(1)$$

where $n_i$ is the number of dosages of drug $i$, $i = 1, \cdots, N_d$. The *Equation 1* accounts for all possible set of drug combinations by including an additional concentration condition $c_i = 0$ for each drug $i$, $i = 1, \cdots, N_d$. For example, the number of all possible combinations of two drugs $A$ and $B$ at a single dose, $N_d = 1$, is $(1 + 1) \cdot (1 + 1) = 4$, namely $\{\emptyset + \emptyset, A + \emptyset, \emptyset + B, A + B\}$. The number of possible drug combinations *Equation 1* scales exponentially with the number of drugs and dosages.

In the case of ex vivo testing, the size of the obtained (tissue or blood) specimen might not be sufficient to probe an enormous dosing parameter space. Another challenging aspect of the experimental identification of optimal combination therapy is logistics. Often, hundreds of samples need to be prepared, cultured, processed, stored, and analyzed. The whole undertaking cannot be entirely robotic, typically requiring human curation to ensure quality control.

## Noise and law of large numbers

Fluctuations in measured biological parameters around their typical values in cellular systems can be described as a consequence of intrinsic and extrinsic noise *Elowitz et al., 2002*; *Raj and van Oudenaarden, 2008*. Here, we will refer to intrinsic noise as phenotype(s) fluctuations across isogenic cell populations cultured under the same conditions *Kaern et al., 2005*. Measurement noise in some cases can also be thought as intrinsic noise. Fluctuations in cellular phenotype(s) driven by the global environment will be referred to as extrinsic noise.

In cell culture, there exist several extrinsic factors that drive heterogeneity in the cells' response to the drug treatment. Among these factors are included cell density, local cell clustering in the well, passage number, time in culture prior to drug exposure, and position on a multi-well plate (leading, for example, to temperature and evaporation rate gradients). All of these factors contributes to extrinsic noise and, paradoxically, the fate of drug response experiments can be (pre-)determined (in terms of fluctuations in the outcome) even before they begin as a consequence. Hence, quantification of drug effects requires multiple replicates, and very carefully designed protocols and handling of the biological specimens.

There also exist many microscopic parameters that drive intrinsic noise in drug uptake. The drug diffusion coefficient, cell membrane potential and capacitance, active transport properties, and intracellular metabolism/degradation of the drug are among parameters controlling drug uptake by individual cells. Due to differences in cell shape and size, cell cycle state, and the microenvironment, these parameters can vary from cell to cell, further driving intrinsic noise in drug uptake.

While intrinsic cell variability can be significant (with coefficients of variation of the order $CV \sim 1$), we believe that it is extrinsic factors that drive overall replicate sample variability in typical cell biological experiments. This conclusion is a consequence of the Law of Large Numbers *Feller, 1968*, since the typical number of cells in a culture sample is at least of the order of $10^4$. According to the law, the average of the results obtained from a large number of trials should be close to the expected value and will tend to become closer to the expected value as more trials are performed. Deviation from the expected value scales inversely proportionally to the square root of the number of trials. Hence, even if intrinsic cell variability is of the order $CV \sim 1$, averaging over 100 randomly chosen cells will decrease this variability to $CV \sim 0.1$, corresponding to 90% noise suppression. By contrast, the typical number of replicates in most biological applications is 3, averaging over which will only suppress noise by roughly 40%.

Throughout this paper, we utilize the Law of Large Numbers in different contexts, and, hence, it is worth demonstrating its action in a concrete biological example. To illustrate the application of the law, we assessed a few commonly used cellular characteristics, such as a gene expression level and dye uptake, at the single cell level. We also measured uptake by individual cells of such different fluorescent species as the Hoechst bisbenzimidine nuclear stain or quantum dot fluorescent nanoparticles (QDs). Detectable fluorescence signals from these agents can be thought of as a correlate to drug delivery: Fluorescence intensity of transient GFP-expressing cells is a direct consequence of Lipofectamine transfection of the corresponding plasmid. Hoechst dye becomes fluorescent only upon binding to the minor groove of DNA. QDs (functionalized by cell penetrating peptides) can be designed to carry a real drug payload. A large degree of intracellular heterogeneity in uptake is shown in *Appendix 1—figure 1a* for these diverse fluorescent species with corresponding reagents delivered sequentially to a single HeLa cells sample (cultured on a microscopy slide).

Using single-cell data as a starting point, we randomized cell ordering and uniformly partitioned data into bins. Each bin contains an equal number of elements, $r$, corresponding to measured parameters (attributes) of $r$ distinct cells. We will refer to $r$ as a coarse graining or scaling factor, borrowing the terms from the statistical physics approach to model reduction (renormalization) developed in 1960–1970s by Kadanoff, Wilson, and Fisher (*Cardy, 1996*). We averaged cellular attributes (e.g. GFP fluorescence) over each bin to produce a renormalized distribution of the phenotype for different values of the parameter $r$ (*Appendix 1—figure 1b*).

A reduction in noise for all three phenotypes was observed after distribution renormalization, as shown in *Appendix 1—figure 1b* where narrowing of the distribution of Hoechst dye fluorescence intensity is demonstrated as a function of the scaling factor $r$ (similar results hold for the other reporters, GFP and QDs). As expected, the degree of variability behaves precisely according to the Law of Large Numbers, as shown in *Appendix 1—figure 1c* where the coefficient of variation ($CV$) is shown as a function of $r$.

While it is reassuring that the Law of Large Numbers works flawlessly in a single cell statistical ensemble, we next need to consider its effect on a range of drug combinations. It is rather obvious that the renormalization process applied uniformly (i.e. randomly choosing cells for each bin) to a cell population results in a reduction of possible combinations precisely as a consequence of this law. This effect is illustrated in *Appendix 1—figure 2a* , where we show a scatter plot of QD and GFP cellular fluorescence intensities before (red) and after (blue) a uniform renormalization procedure.

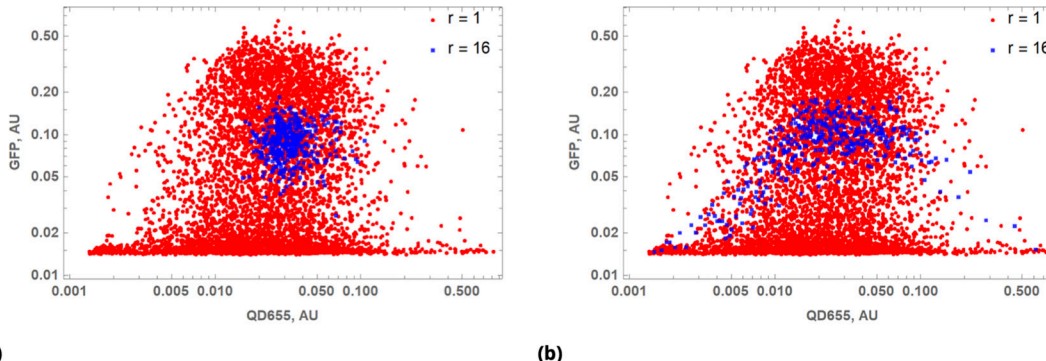

**Appendix 1—figure 2.** Results of multivariable data coarse-graining depend on pre-processing step(s). (**a**) Scatter plot of $QD_{655}$ and GFP cellular fluorescence intensities for raw ($r = 1$) data and *uniformly* renormalized ($r = 16$) data shown in red and blue, respectively. (**b**) The same as in (**a**), but renormalization was performed after sorting cells in ascending order by QD cellular fluorescence intensity.

We can also apply renormalization to sub-populations of cells sharing a common characteristic(s). A trivial example of this process is illustrated in *Appendix 1—figure 3*, where we choose as a common characteristic a color. For example, we can partition the entire population of cells into bins designed to span the whole range of a particular reporter such as Hoechst dye fluorescence intensity. The partition into $N$ bins is given by a sequence of intervals $[b_{i-1}, b_i]$, $i = 1, \cdots N$ covering the entire range $[b_0, b_N]$ of Hoechst dye fluorescence intensity, where $b_0$ and $b_N$ correspond to the minimal and maximal fluorescence intensity values, respectively, for the entire cell population. The resulting partition bins do not contain cells typical for an entire population but, rather, are organized by the degree of closeness with respect to the reporter used for cell sorting. The partitioning for each bin in this way is analogous to gating used in flow cytometry. Hence, we will refer to this partition strategy and subsequent coarse graining as a gated renormalization process (cf., uniform renormalization by binning cells randomly).

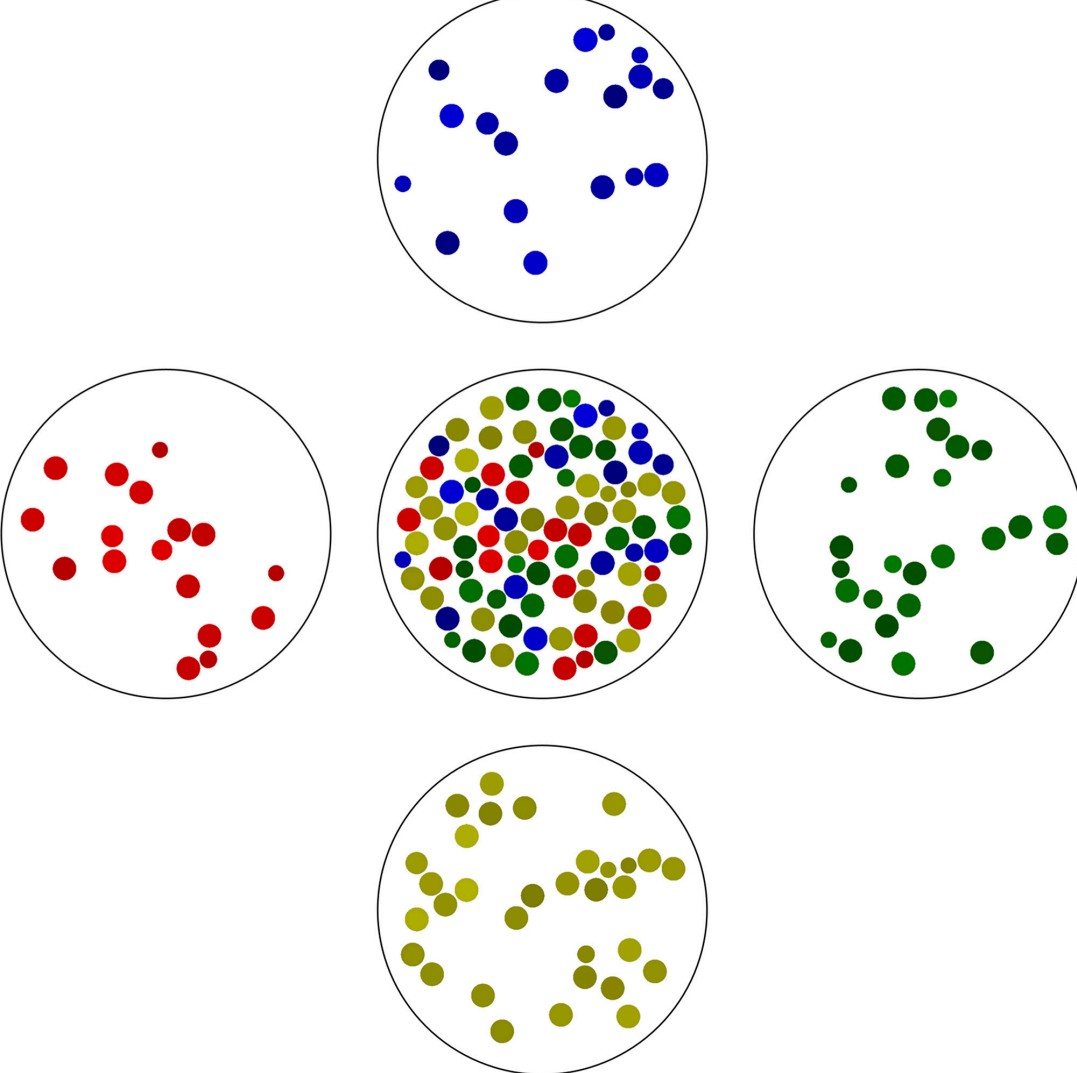

**Appendix 1—figure 3.** The simplest example of gated partitioning is based on the color of the objects. Here, we partitioned all objects shown in the center into a one-dimensional grid with 4 bins spanning the color spectrum. Note the variation in 'cell size' in each bin.

Gated renormalization is designed to integrate out fluctuations in attribute(s) of the cell that are not involved in the partitioning criteria. As a trivial example shown in *Appendix 1—figure 3*, the 'cellular' attribute not involved in the clustering and partitioning is 'cell' size, which has large fluctuations among each partition bin in this example. By contrast, the gating parameter(s) for each bin is(are) expected to lie somewhere within the gating criteria $[b_{i-1}, b_i]$ for each bin $i$ after averaging. If cell attributes are all statistically independent, gated renormalization will result in coarse grained, but still statistically independent, attributes; however, if some attributes correlate with one another, the gated renormalization is expected to make this dependence more apparent by reducing biological and measurement noise.

The variance across data renormalized in this fashion is expected to be comparable to that of raw data for gating attributes by construction. The variance in the cellular attributes not involved in the partitioning criteria after gated renormalization will depend on the degree of correlation between these attributes and the gating parameters(s). As an example of this point, in *Appendix 1—figure 2b* we show a scatter plot of QD and GFP cellular intensities with renormalization performed after sorting in ascending order of fluorescence intensity. This strategy can be generalized to design mesoscopic partitioning of a cell population into multi-dimensional bins characterized by a range of multiple reporters (e.g. by choosing Hoechst and GFP fluorescence intensities as gating parameters).

The gated partition data analysis for identification of the optimal drug combinations is, then, rather straightforward. First, we rely on natural biophysical phenomena to provide a broad range of random mixing in drug uptake. Next, we use the fluorescence intensity of drugs (or their corresponding tags) to partition all cellular attributes into bins. These bins are designed to be sufficiently capacious to contain dozens or hundreds of cells. All cellular attributes are averaged within these bins, which map out the space of drug concentrations and corresponding cellular responses/phenotypes. Finally, we treat data from each bin as an individual culture dish with delivered drug concentrations proportional to fluorescence intensities of the corresponding tags. (According to the Law of Large Numbers, averaging drug response over many cells in each bin suppresses intrinsic noise by the factor $\sqrt{n_i}$, where $n_i$ is the number of cells in bin $i$).

## Point injection and conjugation-free method

We labeled each drug uniquely with fluorescent tags, and assumed that the amount of drug taken up by each cell scales as the fluorescence of its corresponding tag. The quantification of each cell response is matched to fluorescence data by utilizing tags that are retained in cells. The inherited randomness in cell behavior is not only responsible for variable drug uptake, but also drives response to drug treatment by individual cells. To overcome this source of heterogeneity, we perform averaging over many cells within the sub-population binned by fluorescence intensities of drug tags.

There are potential pitfalls in the random mixing method. For example, the uptake of different drugs may be correlated due to the similarity in chemical structures or delivery methods. Cells that take up one drug may, therefore, be likely to take up a similar drug, as well, in proportional fashion. Hence, drug mixing becomes not very effective since we are not exploring the entire drug mixing range by introducing a bias in the mixing process.

Multiple approaches can potentially mitigate this "proportionality" bias. One possibility is to utilize different delivery mechanisms for different drugs. This approach, however, is too cumbersome and does not guarantee bias elimination. Randomness in drug uptake might be driven by similar factors (such as membrane geometry and physical properties), even for different delivery methods.

Here, we developed a method to facilitate independent drug mixing by the creation of local drug gradients across the sample using point injection of individual drugs in different sample locations. The modification of the point injection method can be used for conjugation-free drug labeling by local co-delivery of a pre-mixed drug and its corresponding label. We confirmed that the conjugation-free drug labeling method works well, not only with dyes or small molecule drugs, such as auranofin, but also with much larger nanoparticles, such as QDs.

To assess the predictive power of the conjugation-free method, we compared the CV for each cell subpopulation (bin) used to produce the coarse-grained data shown in Section 6 and Section 6 to the CV of the entire population in a uniform dye delivery experiment. The results of this comparison are shown in *Appendix 1—figure 4a* and *Appendix 1—figure 4b* for dyes we treated as 'drugs'. Here, the large red circles represent the CV of the entire cell population in the uniform delivery method for these drugs.

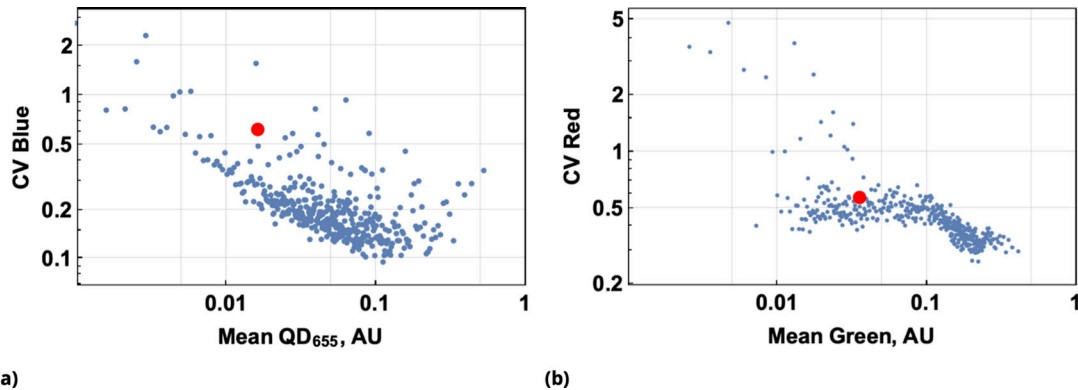

(a)                                                    (b)

**Appendix 1—figure 4.** Predictive accuracy in drug uptake using the point co-delivery method. (**a**) and (**b**), Predictive accuracy in drug uptake using the point co-delivery method; the coefficient of variation in 'drug' uptake for each bin of co-delivered dyes is shown for pairs {Blue, QD$_{655}$} and {Green, Red}, respectively. Here, the
*Appendix 1—figure 4 continued on next page*

*Appendix 1—figure 4 continued*
CV of 'drug' uptake is plotted against the average dye uptake for each bin comprised of 100 cells characterized by similar dye intensities. Variability in the uniform 'drug'-dye co-delivery experiment across the entire cell population is shown as the large, filled, red circles for each condition above.

## Coarse-grain image processing by tiling

In order to utilize stochasticity in drug uptake (or any other cellular phenotype) as a tool to study biological correlations, one has to have enough cells to make statistical analysis feasible. This requirement is essential for two primary reasons. First, a large statistical ensemble allows one to collect enough information about atypical (rare) cellular responses to drug treatment that may nevertheless be critically important (e.g. resistance to treatment). Second, renormalization requires partitioning and subsequent bin averaging to suppress intrinsic noise, which is best achieved using a large ensemble.

While flow cytometry-based experiments can easily be designed to handle large numbers of cells, this is usually not the case for fluorescence microscopy. The experimental protocols, data storage, and analysis are usually non-trivial technically. Confluent cell culture conditions can present a problem for segmentation analysis both in terms of accuracy and run-time. Hence, many tiled optical fields might be necessary to image a large number of cells. Cell segmentation may also require a dedicated optically active biomarker resulting in bandwidth reduction and possible leakage to other channels in high-throughput experiments designed to assess multiple cellular attributes.

To overcome these issues with confocal fluorescence microscopy methodologies, we employed yet another coarse graining method for image analysis. Similar to renormalization of the cell population, microscopy images are renormalized as follows. All channel images are identically partitioned into square tiles of an area large enough to contain at least one cell on average. Fluorescence intensities for each of these tiles are averaged, and the resulting mean intensities are treated as a single-cell attribute. Of course, the data collected in this fashion would include intensities from incomplete cells, fragmented by the tile limits, and also intensities from multiple cells averaged together.

To study these effects on tiled renormalization and assess its quality, we generated synthetic images of geometrically regularized but dense objects (circles), as shown in *Appendix 1—figure 3*. (*Appendix 1—figure 5a*) or varying radii (*Appendix 1—figure 5b*). We assigned to each circle a unique pixel intensity (uniformly across each circle). The images were partitioned into tiles (we will refer to its tile size as a scaling parameter) as illustrated in *Appendix 1—figure 5b* by the red lines. We calculated average pixel intensity by normalizing total intensity to the foreground area for each tile. To this end, we created a binary image (mask) by separating foreground (circles) and background in the image. Unlike traditional quantification approaches that rely on identification of individual objects and involve separation of touching objects in the image, we used only masks for tile normalization.

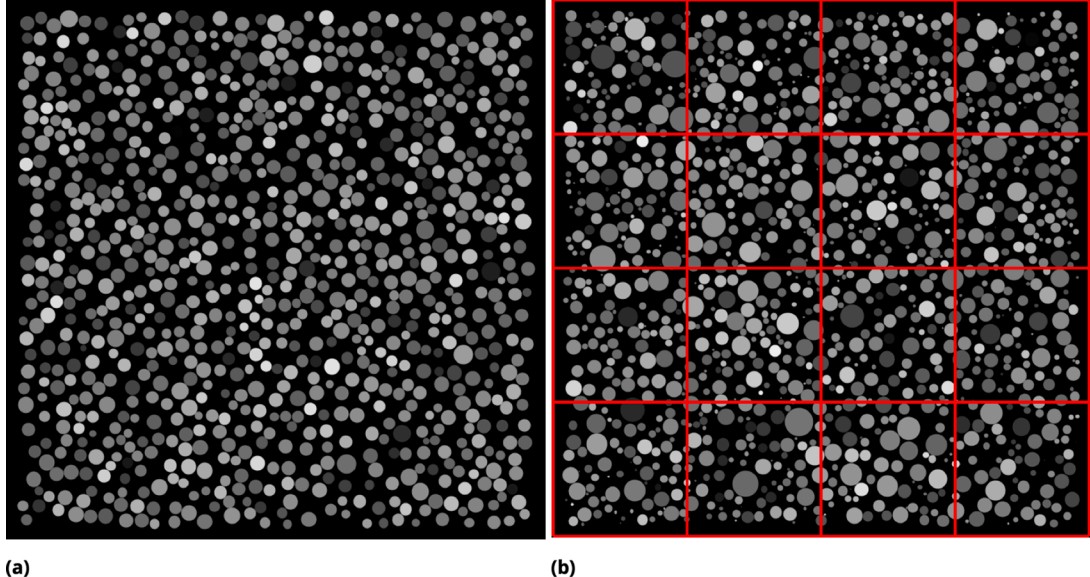

**(a)**                                                     **(b)**

**Appendix 1—figure 5.** Synthetic images used to infer the optimal partition strategy for coarse-graining.
(a) Synthetic image generated by seeding identical circles at random locations in $1000 \times 1000$ pixels square
and assigning random pixel intensity *uniformly* across each circle. Dense culture conditions are emulated by
allowing circles to touch. (b) Circles of varying radius $R$ were generated [radii are drawn from a normal distribution
($\mu_R \sim 10$, $\sigma_R \sim 3.8$ pixels)]. Pixel intensities were assigned as in (a). The concept of tile partitioning is illustrated
by red mesh lines.

The pixel intensities for each circle were drawn from a statistical distribution $\mathcal{D}_e$ (either
uniform or normal). We compared any 'measured' intensity distribution $\mathcal{D}_m$ (either by means of
tiled renormalization or traditional segmentation) with the true/exact function $\mathcal{D}_e$. To this end, we
estimated numerically the overlap between $\mathcal{D}_m$ and $\mathcal{D}_e$ using the earth mover's distance metric
(first Wasserstein/Mallows distance). In our implementation we are simply measuring the area under
the overlap curve and comparing it to the area under the $\mathcal{D}_e$ curve as illustrated in *Appendix 1—
figure 6a*. The overlap value varies from 0 to 1 which corresponds to complete distribution
mismatch or complete match, respectively. The results of a comparison between the measured
and true distributions are shown in *Appendix 1—figure 6b* for different statistical properties of
pixel intensities and circle radii. Here, the dependence on the width of the distribution is more
pronounced for pixel intensity statistics compared to radii of one.

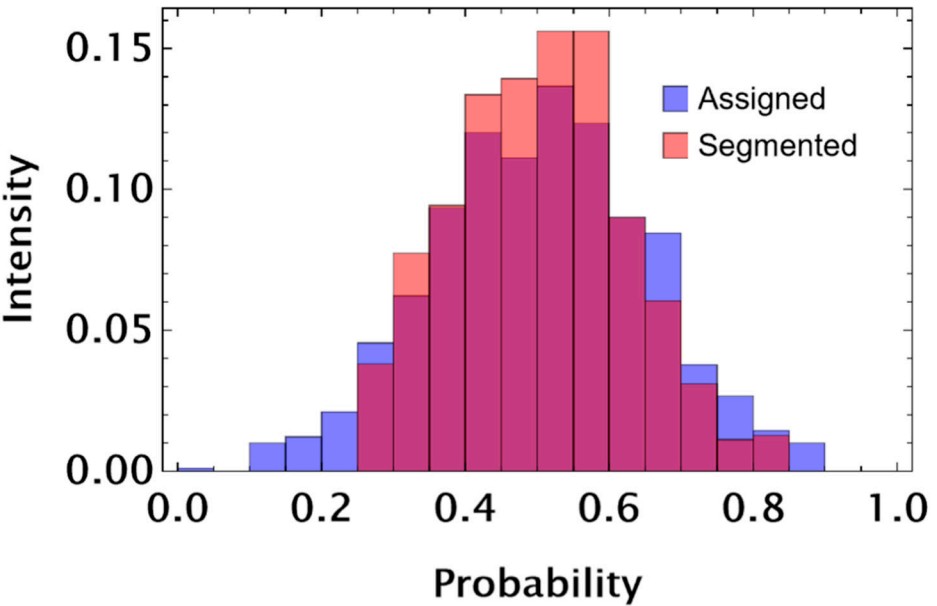

(a)

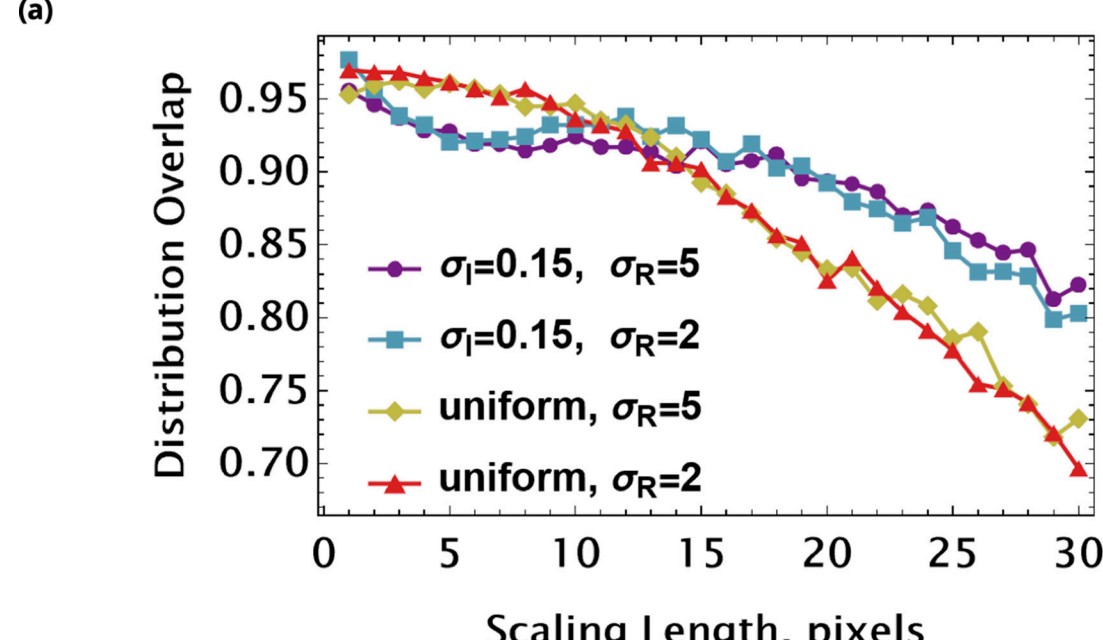

(b)

**Appendix 1—figure 6.** Assessing coarse-graining algorithm performance using synthetic image data (uniform pixel intensity in each cell). (**a**) Graphic illustration of distribution overlap assessment. We compare the assigned (exact) distribution of mean object intensities (blue) with a 'measured' (orange) intensity histogram. The area under the curve is 1 for any probability distribution. Hence, the area of overlap (purple) can be used as a dimensionless parameter reflecting a match between exact and measured distributions. 'Measured' data here are derived using Otsu's global thresholding (image segmentation) method. (**b**) The distribution overlap values as a function of scaling length used to partition synthetic images. The different curves correspond to different statistics of pixel intensities and circle radius. In all conditions, the mean values of pixel intensity and radius are $0.5AU$ and $\sim 10$ pixels, respectively. The pixel intensities were drawn from either a uniform or a normal distribution characterized by $\sigma_I = 0.15$. The radii of circles were drawn from a normal distribution with $\sigma_R = 1.5$ or $\sigma_R = 3.8$ pixels.

To explain why a single pixel partition leads to the best match between renormalized and exact distributions, consider the case of identically sized objects, $A$ pixels each. In this case, each object's

contribution to the overall single pixel distribution will be $A$ identical values $I_k$ where $I_k$ is an intensity assigned to the object $k$. While $I_k$ varies from object to object, pixel size $A$ is constant. Hence, a single pixel distribution should be identical to that for objects' intensities $\{ I_k \}$, $k = 1, 2, \cdots$. (Histogram binning may slightly affect the perfect match with an overlap of 1.) Therefore, we expect that as long as fluctuation in object intensity dominates variations in object size, the dependence of the distribution overlap on scaling length should monotonically decrease. This intuition, however, only holds for uniform objects in the image. Here, we synthesized more realistic images by introducing noise across individual circles' intensities. We applied a multiplicative noise function, $G_\phi(I)$, for each pixel of value $I$, as follows

$$I \to G_\phi(I) = (1 + \phi\, \zeta)I, \qquad (2)$$

where a random variable $\zeta$ is drawn from a uniform distribution (for example), and the strength of noise is controlled by parameter $\phi$.

The results of these simulations are shown in *Appendix 1—figure 3*. The typical synthetic image and its traditional segmentation are shown in *Appendix 1—figure 7a*. Here, we used the following statistical parameters to create this image:

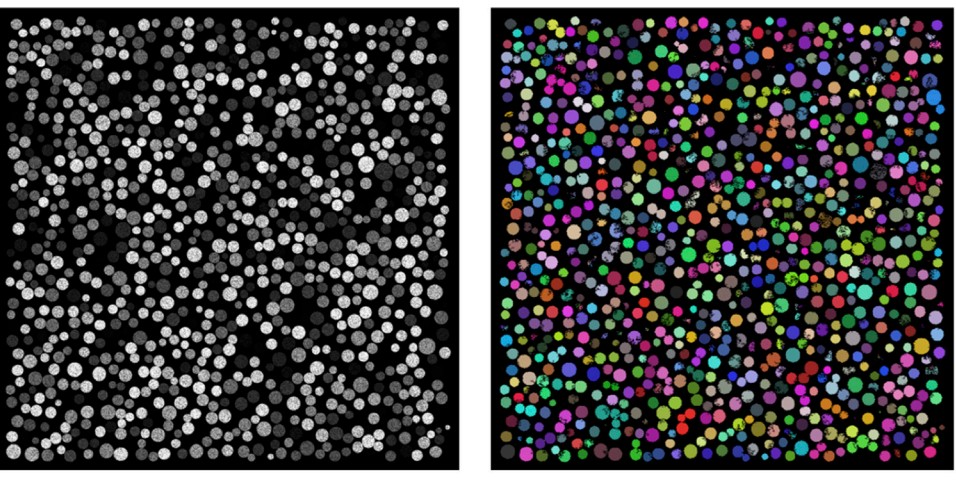

**(a)**

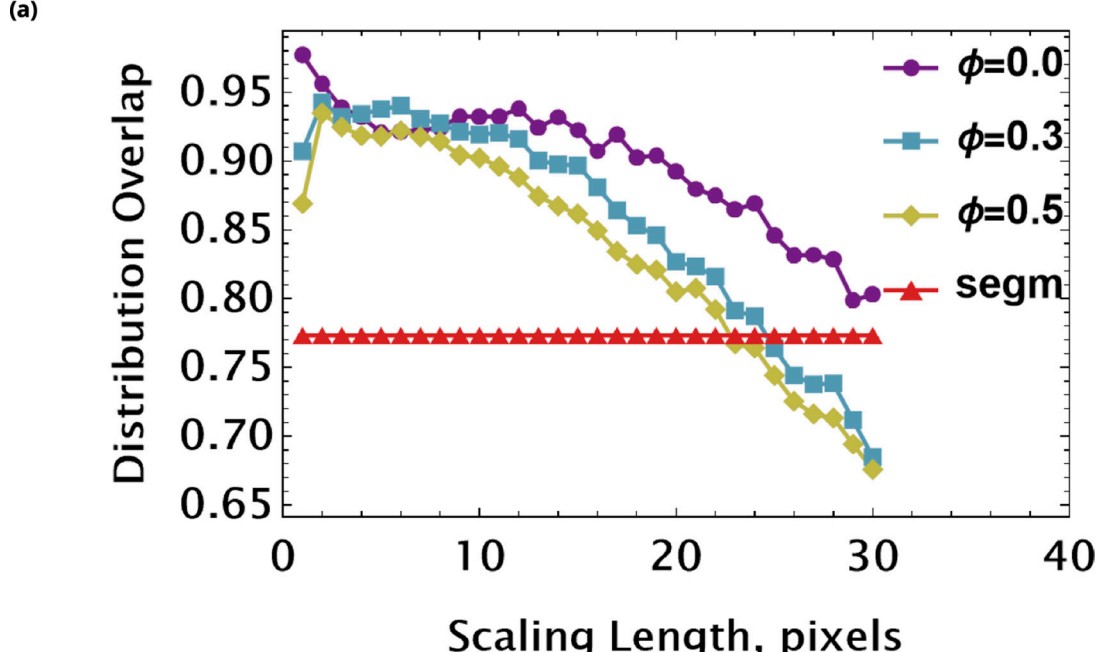

**(b)**

**Appendix 1—figure 7.** Assessing coarse-graining algorithm performance using synthetic image data (non-uniform pixel intensity in each cell). (**a**) Non-uniform, noisy pixel intensities across each circle were simulated by introduction of multiplicative noise. The synthetic image (left) was segmented using a local, adaptive binarization approach. Imperfections of segmentation due to noise and touching objects are shown in the colorized binary image (right). (**b**) The overlap between exact and measured (using tiling renormalization) distributions are shown as a function of scaling length. Different curves are characterized by the strength of noise controlled by parameter $\Phi$. The flat red line is an overlap value between exact and data-derived distributions using a standard segmentation approach (**a**).

$$\{\mu_R \sim 10, \sigma_R \sim 1.5\} \text{ pixels},$$

$$\{\mu_I = 0.5, \sigma_I = 0.15\},$$

$$\phi = 0.5.$$

The comparison of the measured and exact intensity distributions' overlap as a function of scaling parameter is shown in *Appendix 1—figure 7b* for different values of noise strength $\phi$. The red flat line corresponds to the overlap value obtained by comparison of segmented and exact distributions' overlap for $\phi = 0.5$. We used local adaptive segmentation and a smoothing approach to obtain this value.

It is clear from *Appendix 1—figure 7b* that unlike the case of noise-less (uniform) objects, tile partitioning may have an optimal scaling parameter value. This conclusion is because a consequence of the fact that a single object in the image does not contribute A identical values to the overall distribution in this case. Averaging over finite sized regions suppresses multiplicative noise in the model we introduced. Naively, we expect similar behavior in real microscopy data.

We found that the tile renormalization (and segmentation-free) approach, indeed, results in fairly accurate estimates of the statistical properties of cellular attributes (see *Appendix 1—figure 3* for comparison between segmented single cell and renormalized [tiled] image statistics for the phenotypes we tested). We must note that, in principle, accurately determined statistics of each individual random variable in the multidimensional ensemble does not necessarily guarantee a correlative relationship between these variables. For example, some cellular markers (such as QDs) tend to be localized in sub-cellular compartments, and too aggressive image partitioning (i.e. too small a scaling parameter) may result in loss of the correct relationship between these localized markers. By contrast, too large a scaling parameter results in loss of sensitivity and range in the determination of the quantitative relationship between the markers. We, therefore, relied on a comparison of segmented single cell and tiled data to determine the optimal trade-off in the partition strategy empirically (using sparsely seeded control samples where traditional segmentation is possible).

For confluent cell culture conditions (or tissue), the application of tiling renormalization is straightforward. However, when lacunae are present in the culture due either to seeding inhomogeneities or cell death, one has to use masking to prevent averaging over empty space. Masking does not require segmentation but, nevertheless, may require an additional fluorescence marker or phase-contrast imaging to define the approximate limits of the lacunae.

## siRNA delivery by quantum dots nano-carriers

After conjugation of streptavidin-QDs with biotinylated siRNA, we further functionalized QDs with an arginine-rich cell-penetrating peptide (CPP) that facilitate cell entry *Medintz et al., 2008*. The peptide, biotin-labeled (Arg)$_9$, was purchased from Anaspec (AS-64078).

As in the case of the diester dye experiments described above, we observed bias towards proportional drug uptake when the QD nanocarriers were delivered uniformly across samples (*Appendix 1—figure 9a* and *Appendix 1—figure 9b*), which is, almost certainly, a consequence of identical drug delivery and uptake mechanisms. All of our drugs were siRNAs, and all of them were delivered in exactly the same fashion via QD nanoparticles functionalized by the same CPP. Hence, one expects the uptake kinetics of each drug to be quite similar for each individual cell (the kinetics, of course, varies widely among different cells in the population). Point injection of QD-conjugated siRNAs produced a concentration gradient that resulted in much broader range of drug combinations across the samples (*Appendix 1—figure 10*).

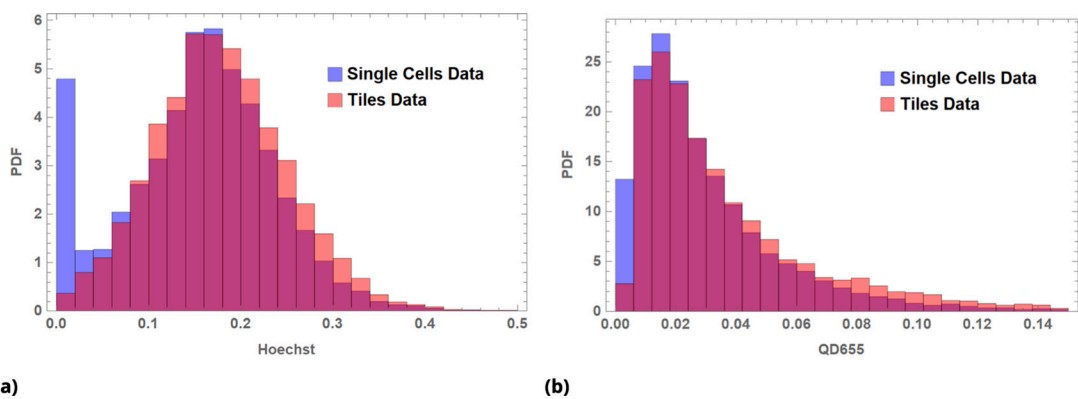

(a)　　　　　　　　　　　　　　　　　　　(b)

**Appendix 1—figure 8.** Image data renormalized by tiling for (**a**) Hoechst dye and (**b**) QD$_{655}$ fluorescence intensity data. Blue, single cell data; orange, tile data; red, overlap.

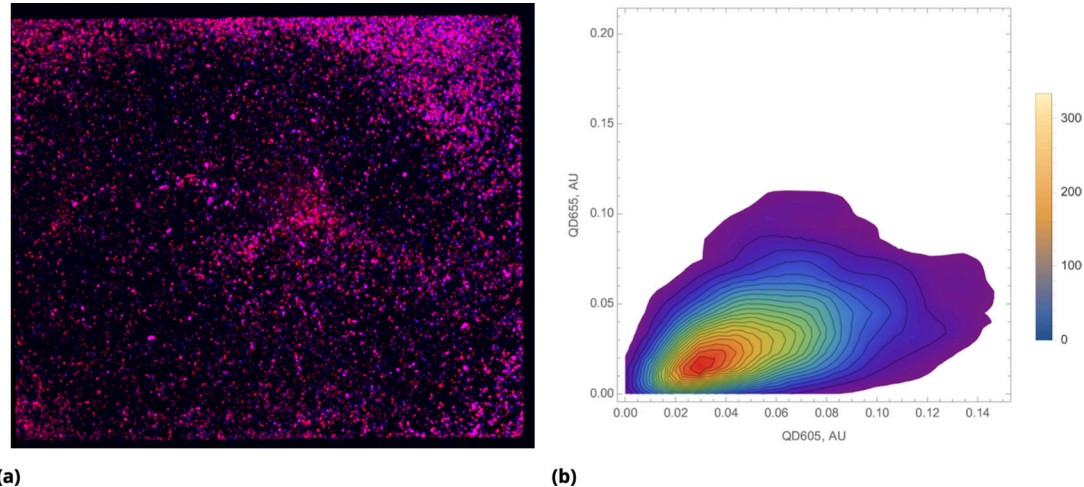

(a)　　　　　　　　　　　　　　　　　　　(b)

**Appendix 1—figure 9.** Bias towards proportional drug uptake in distribution of pre-mixed drugs. (**a**) Color combined image data of the entire cell culture plate (chamber slide) after incubation with pre-mixed QD$_{605}$ and QD$_{655}$ nanoparticles delivered uniformly (shown in blue and red, respectively). (**b**) Density plot of renormalized QDs' intensities for uniform delivery.

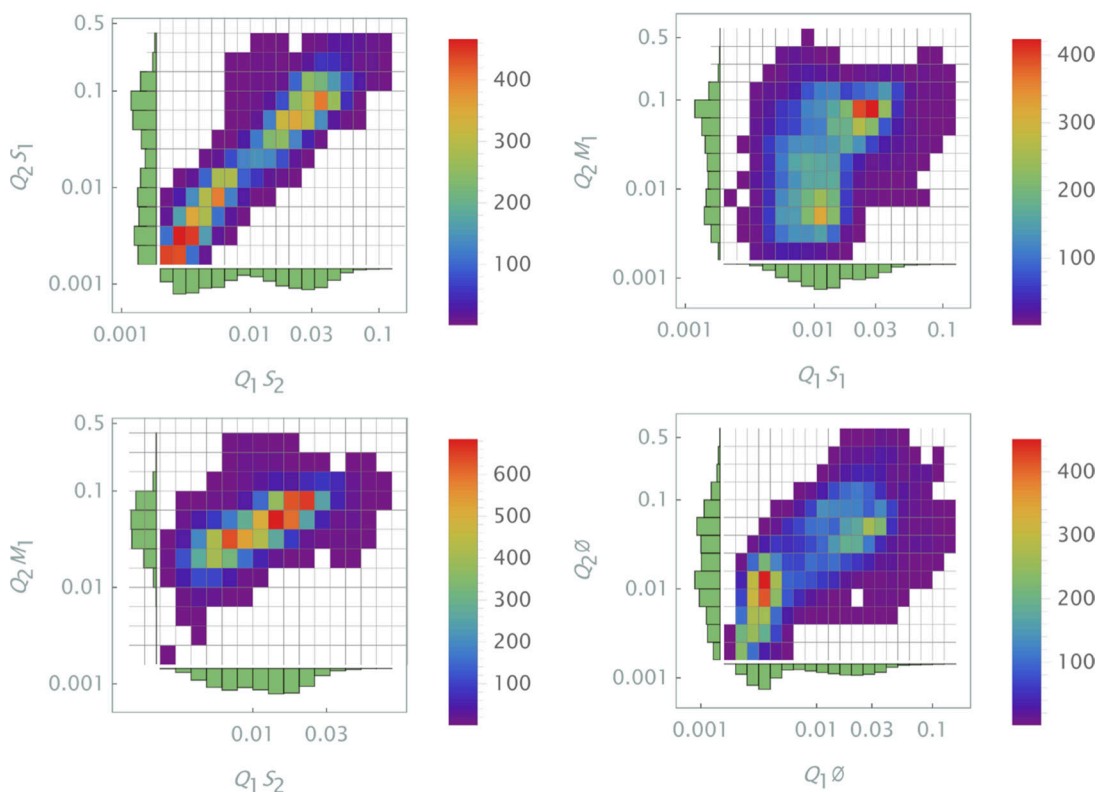

**Appendix 1—figure 10.** Density plot (log scale) of renormalized QDs' intensities for point delivery of several representative samples. Different siRNA combinations were delivered pair-wise by means of functionalized quantum dots, $QD_{605}$ and $QD_{655}$.

## Bin partitioning

We first assign to each cell $n$ a set of $d$ measured attributes

$$\vec{x}^n = \{x_1, x_2, \cdots, x_d\}^n \tag{3}$$

where $n$ is an index denoted by a superscript, and the subscript is used to distinguish attributes. The bin partitioning is performed with respect to a subset of the attributes. To be specific, let us define the partition and subsequent subpopulation average based on a first attribute, $x_1$.

$$\langle \vec{x} \rangle_{x_1} = \langle \{x_1, x_2, \cdots, x_d\}_n \rangle_{x_1 \subset [x_1, x_1 + \Delta x_1]} \tag{4}$$

The generalization to a higher dimension is straightforward. For example, for pairwise drug combinations, one has to perform binning and averaging over attributes $\{x_i, x_j\}$ simultaneously satisfying conditions

$$\{x_i, x_j\} = \begin{cases} x_i \subset [x_i, x_i + \Delta x_i] \\ x_j \subset [x_j, x_j + \Delta x_j] \end{cases} \tag{5}$$

To decide what mesh sizes $\Delta x_i$ should be used for each attribute $i, 1, \cdots d$, we applied the following strategy. We set as a goal to partition the underlying parameter space into the mesh in such way that each bin contains a similar number of cells. The average number of cells, $L_d$, in each bin is pre-determined from considerations of noise suppression. The number $L_d$ depends on the width of the joint distribution of $d$ attributes comprising the parameter space; we typically chose $L_d \sim 30–100$ for single-cell data, and $L_d \sim 3–10$ for tiled image data.

To find a space partition mesh satisfying these criteria, we developed the following approximation. First, we assumed that binning and averaging are performed over *independent* attributes, such as

QD carriers delivered independently by point injections. This very strong assumption reduces the problem into finding a partition in 1d space for each attribute.

For each attribute, $x$, we quantile the intensity distribution of collected $N$ values of attribute $x$ to ensure equal number of events, $L$, for each bin. If our assumption holds that all $d$ attributes $x$ are statistically independent, the number of events $L_d$ in a $d$-dimensional partition mesh is given by:

$$L_d = N \left( \frac{L}{N} \right)^d \tag{6}$$

Therefore, an estimate for the value $L$ we must use in 1d partitioning is:

$$L = N^{(d-1)/d}(L_d)^{1/d} \tag{7}$$

For example, for $L_d = 100$ and $N = 10^4$, the 1d partition should incorporate $L \sim 10^3$ events per bin. We optimized the partition scheme for one particular specimen, namely, the control sample. (Alternatively, all sample data can be pooled, and partitioning can be optimized for the pooled intensity data.) We used this strategy because all samples were normalized to control data and, at least intuitively, we believe that mesh design should perform best for this sample.

In *Appendix 1—figure 11*, we demonstrate the 1d partition outcome for the same partition scheme applied to seven collected samples simultaneously. Here, we display event numbers $L$ in each bin. Optimal performance corresponds to a line with zero slope (equal event numbers in each partition bin, $L = const$). It is entirely possible to find an optimal bin partitioning (mesh) in 2d directly, but we found fairly satisfying results in terms of the number of events per bin using the attribute independence assumption (*Appendix 1—figure 12*).

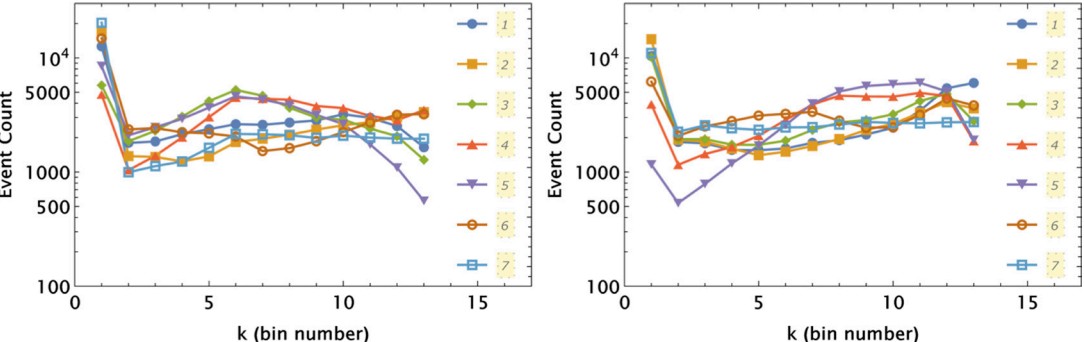

**Appendix 1—figure 11.** Independent application of the 1d partition mesh to $QD_{605}$ and $QD_{655}$ intensity data collected in 7 different samples shown by different symbols and line styles, as indicated in the left and right panels, respectively. The partitioning was optimized for a control sample (7) and applied uniformly to all seven samples.

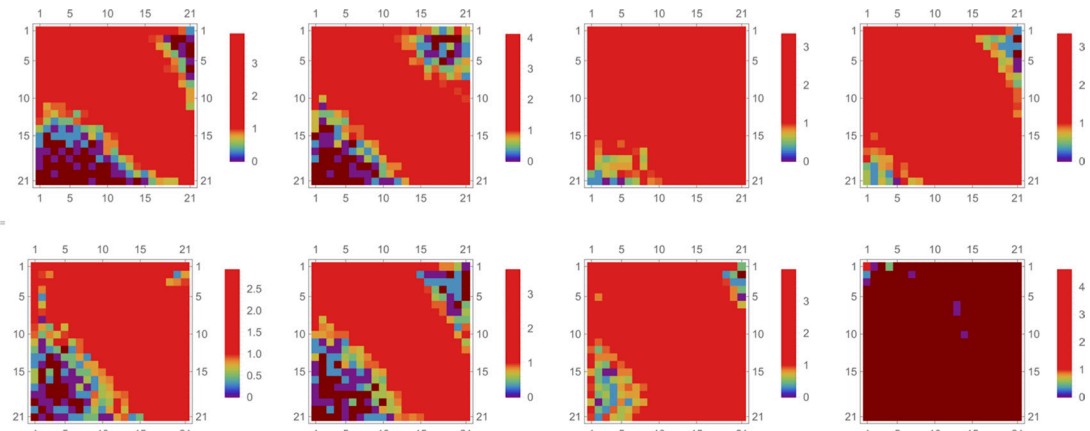

**Appendix 1—figure 12.** Logarithm of event number, $\log_{10}(L_2)$ for a 2d partition mesh of $\{QD_{605}, QD_{655}\}$ intensity data is shown by color (brown color corresponds to no events). Each attribute (both QD intensities) is partitioned independently into 13 bins, creating a mesh of 21x21 bins in 2d space. Here, we applied the same partitioning mesh to eight different samples. The partitioning was optimized for a control sample (7) and applied uniformly to all eight samples [the last (8th) sample corresponds to a QD- untreated specimen].

## Normalization of GFP expression in siRNA experiments

In all siRNA co-delivery experiments, we normalized GFP expression to that of control samples where corresponding QDs were functionalized with CPP only (i.e. with no siRNA cargo). To match conditions, we used the same QD bin partitioning for all samples, including controls. Each bin was characterized by similar QD combinations and, hence, cells within each bin were expected to have similar degrees of toxicity from QDs.

The results of siRNA combinations on target gene (GFP) expression using this normalization process are shown in *Appendix 1—figure 13*. Here, color corresponds to relative GFP expression, varying from 0 (violet) to 1 (red), with values above 1 saturated in red color. Graphics interpolation was used to depict discrete sampling points defined by our renormalization scheme. The mesh partitioning used for renormalization is shown as a grid on each graph.

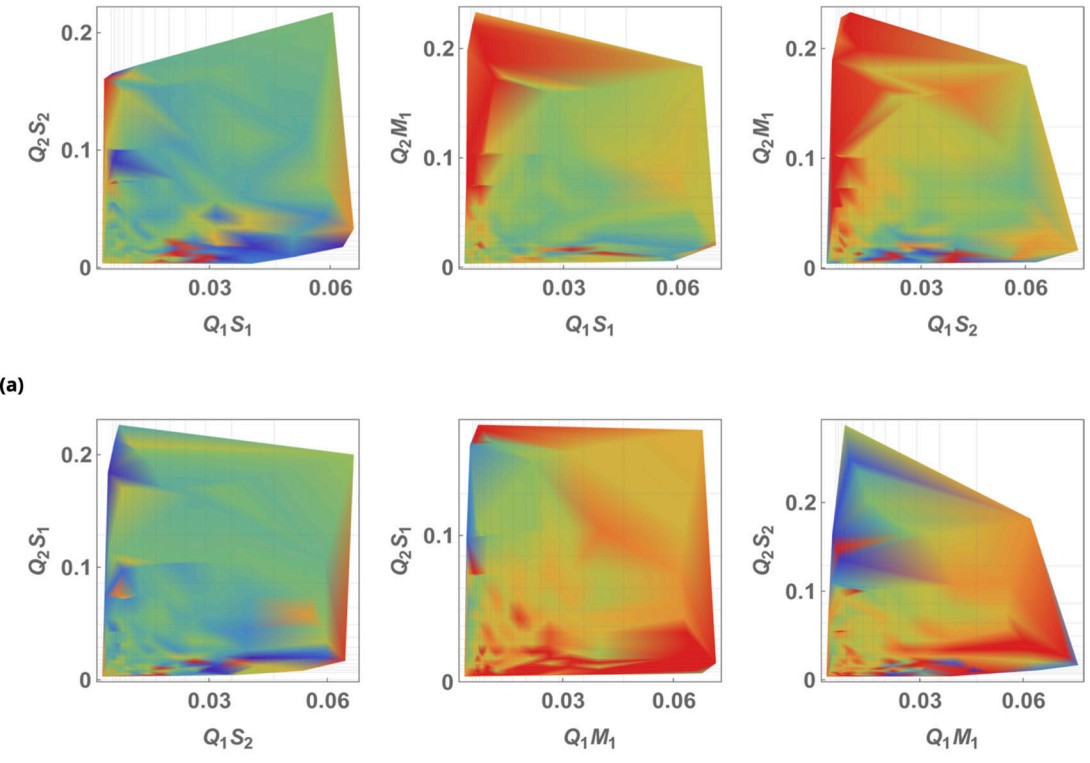

**(a)**

**(b)**

**Appendix 1—figure 13.** The effect of siRNA combinations on target gene (GFP) expression. Three different siRNAs were delivered to cells by optically distinguishable QD nanocarriers ($Q_1$, $Q_2$) in pair-wise fashion. Two siRNAs, siRNA$_1$ ($S_1$) and siRNA$_2$ ($S_2$), were designed to bind non-overlapping sequences of the mRNA target in completely complementary fashion. The third siRNA, miRNA$_1$ ($M_1$), has almost identical primary structure as siRNA$_1$ with a single nucleotide mutation close to the seed region. (**a**) Non-overlapping siRNAs, siRNA$_1$ and siRNA$_2$, were delivered to cells by either QD$_{605}$ or QD$_{655}$ nanocarriers respectively. (**b**) The same experimental setup, but QD carriers are switched for siRNAs.

## Continuous normalization of GFP expression

An alternative strategy that can be used to model the effect of QD toxicity is to employ a machine learning algorithm that is trained on a data subset and validated against a test subset. In the absence of siRNA, QD toxicity was expected to correlate (negatively) with the concentrations of QDs delivered in each individual cell (QD$_{605}$ and QD$_{655}$).

We employed a supervised machine learning predictor $F(Q_1, Q_2)$ based on the $K$-Nearest Neighbors algorithm to estimate GFP dependence on QD fluorescence in an siRNA-free sample. The results are shown in *Appendix 1—figure 14*. Using function $F(Q_1, Q_2)$, we normalized GFP intensities in siRNA samples as shown in the main text.

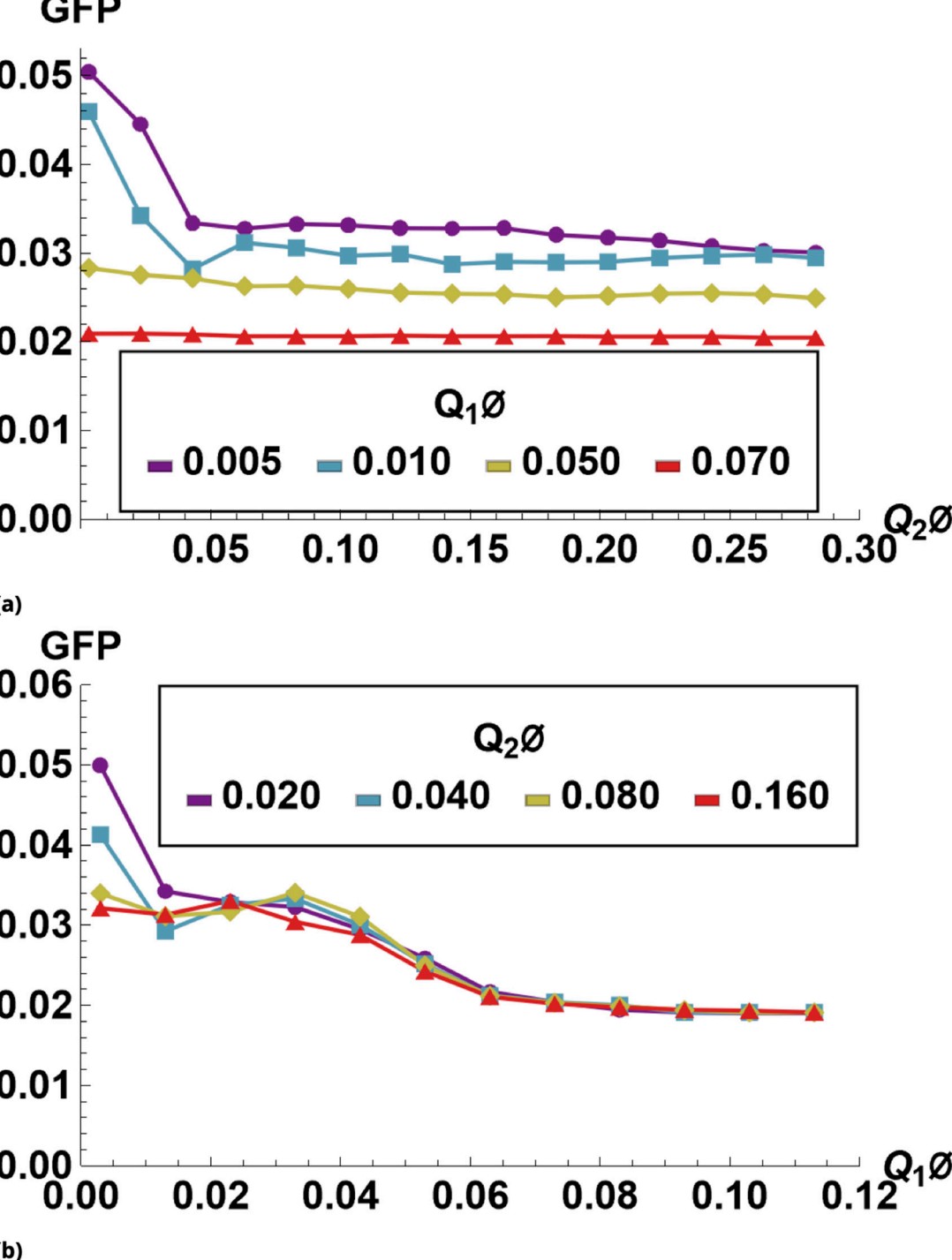

**Appendix 1—figure 14.** Supervised machine learning prediction for GFP inhibition by QDs. We used tiled renormalization applied to siRNA-free control sample images to derive the function $F(Q_1, Q_2)$ that predict GFP expression numerically. Predicted values of GFP expression as a function of either $QD_{655}$ or $QD_{605}$ are shown in (**a**) and (**b**), respectively.

## Compensation of flow cytometry data and identification of intact cells

To distinguish between intact cell and cell debris we utilized forward light scatter signal intensity and width (FSCH and FSC-W) as proxy of cell size. In order to convert fluorescent tag signal to corresponding drug concentration we employed the following normalization scheme.

To eliminate bleeding between channels ('PB450', 'FITC', 'PE', and 'APC') we used data from control samples (single drug/dye). The fitting function for each detection channel was determined after preliminary data processing by moving average filter. The results of the pre-processed data fitting are shown in *Appendix 1—figure 15* for compensation due to CellTracker Deep Red dye used as an homogeneous label dye in our experiment. The leakage between other fluorescent signals was less than 5% and was neglected. The compensation was implemented by utilizing a look-up table.

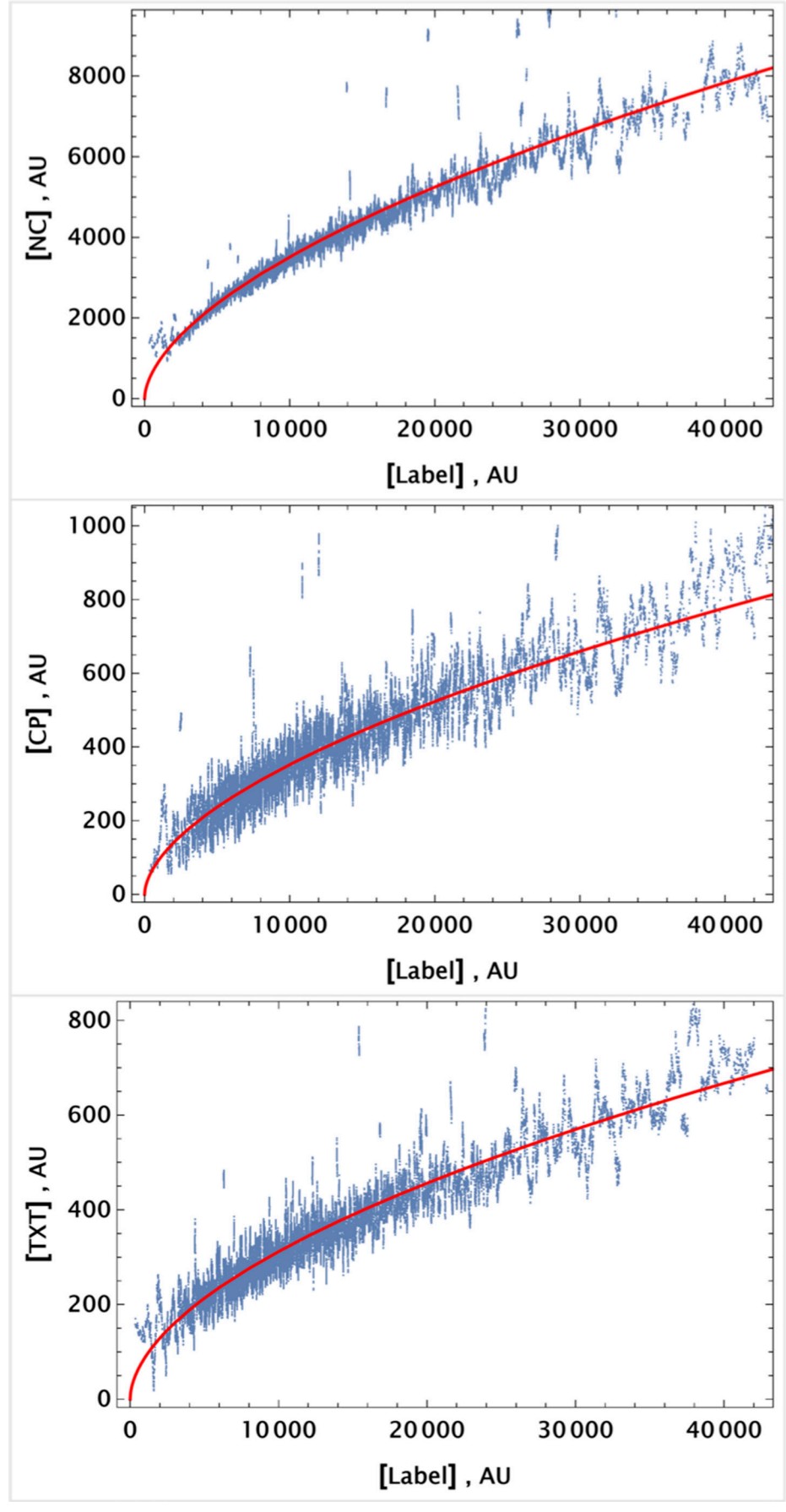

**Appendix 1—figure 15.** Pre-processed data from a control sample (cells labeled with CellTracker Deep Red dye only). Data were sorted using the APC channel signal, and a moving average filter (of depth 50 events) was applied to reduce signal noise. Each channel output was fitted using a simple power law function of the APC (far-red channel) signal. Pre-processed data and the corresponding fit are shown as scattered plots and solid curves, respectively, for each channel.

## Initial spatial distribution of drug concentrations

Characterization of initial spatial distribution of drug concentrations of NC, CP, TXT, and 5-FU is shown in *Appendix 1—figure 16*. We used coarse-grain approach to establish smoothed density distribution for the visualization purposes. The tiled images were stitched together using proprietary ZEN-Blue software. All images were rotated and cropped uniformly (to exclude regions outside of the culture slide). The images were subsequently downsampled by factor $\sim 10$ using piecewise linear interpolation for each fluorescent channel. (Each pixel in downsampled image is roughly equivalent to a single 'cell' in terms of the area.) The images were smoothed using Mean filter with linear window size 10 pixels to average the signal over roughly 100 'cells'. The smooth contour (elevation) plots of estimated drug densities for each of the drugs across entire slide are shown in *Appendix 1—figure 16*.

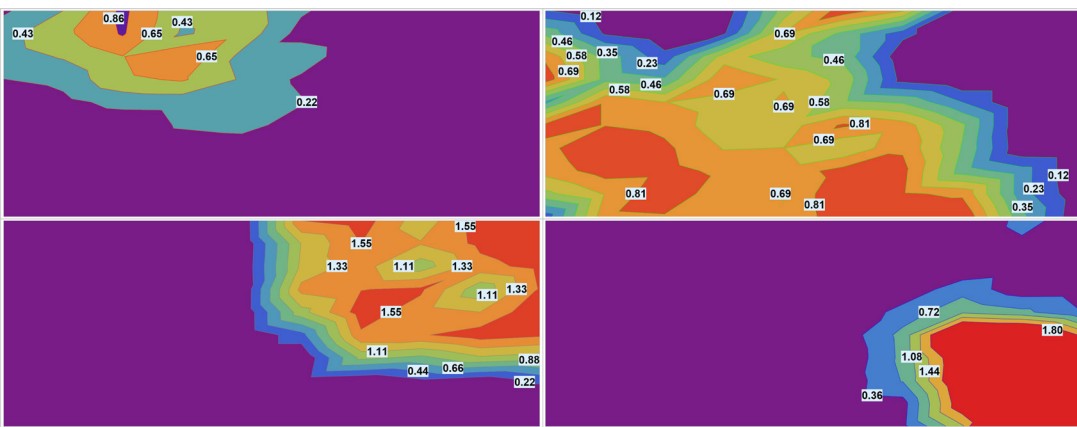

**Appendix 1—figure 16.** Smooth contour (elevation) plots of estimated drug densities for each of the drugs. The images were downsampled by factor $\sim 10$ using piecewise linear interpolation for each fluorescent channel. (Each pixel in downsampled image is roughly equivalent to a single 'cell' in terms of the area.) The images were smoothed using Mean filter with linear window size 10 pixels to average the signal over roughly 100 'cells'.

## Comparison to other combination studies

We note that our method is completely experimental and does not require a priori knowledge of drugs' pharmacokinetics/pharmacodynamics, either individually or in pairwise combinations. Recently, multiple studies employed a mechanism-free quadratic optimization approach that correlates single drug dosages and phenotypic outputs, such as cell death, to predict the effect of drug combinations *Wood et al., 2012*; *Zimmer et al., 2016*; *Zimmer et al., 2017*; *Palmer and Sorger, 2017*; *Menden et al., 2019*; *Palmer et al., 2019*; *Ianevski et al., 2019*. The results are very encouraging since, at the very least, this approach can significantly reduce an enormous parameter space needed for experimental validations. However, in many, if not most, cases, the response to drug combinations in eukaryotic cells is not given by a simple superposition of independent responses to monotherapies *Tekin et al., 2018*; *Diaz et al., 2020*, especially in vivo. Furthermore, theoretical predictions were developed mainly for a specific, profound phenotype, namely, cell death, for both microbiological and human cell culture systems.

