## [Editor Report]

This paper provides a road map for evaluating drug combinations using a single sample. It should be particularly useful for other biologic laboratories to test agents in a manner that does not require major resources, or microfluidics and could reduce costs and time to required for ultimate testing in vivo.

---

## [Decision Letter]

**Decision letter after peer review:**

Thank you for submitting your article "Local Generation and Efficient Evaluation of Numerous Drug Combinations in a Single Sample" for consideration by *eLife*. Your article has been reviewed by 3 peer reviewers, including Clifford J Rosen as the Reviewing Editor and Reviewer #1, and the evaluation has been overseen by Mone Zaidi as the Senior Editor.

The reviewers have discussed their reviews with one another, and the Reviewing Editor has drafted this to help you prepare a revised submission. Overall there is enthusiasm for this novel approach. Please see the notes below:

Essential revisions:

Overall there was enthusiasm for the essential components of the paper and the possible application in other laboratories. There were a couple of concerns that need to be addressed:

1) Have the authors tested other cell lines beyond HeLa cells and other phenotypes beyond cell death?

2) A critical focus is on the cell numbers required for feasibility. Can the authors clarify the numbers of cells that minimize variability.

3) Clarification of "noise" and that external factors that drive significant variability

Other discussion points are noted below for your review.

4) Can the authors clarify how they would control the issue of dye cell leakage for longer-term experiments

*Reviewer #1 (Recommendations for the authors):*

In this paper, the authors develop a method that allows one to test a large number of drug combinations in a single cell culture sample. Their new work is based on their previous studies with drug targeting at the single-cell level. In principle, the experiments rely on randomness of drug uptake in individual cells as a tool to create and encode drug treatments, and for the most part, the experiments validate that approach. The siRNA studies are particularly provocative and provide a novel means of using combinatorial si and i-RNA agents. As such this manuscript has the potential to change how investigators test combinatorial treatments with a single-cell approach. Nevertheless, there are several concerns that should be addressed through other experiments or further discussion of the limitations beyond the issue of random mixing of drugs.

First, did the authors use other cell lines to see if their approach is applicable, especially in non-malignant primary cells or distinct cell lines.

Second, much of the application success is based on the assumption that there is minimal leakage across cells of the fluorescent tags, and that whatever does leak is redistributed in a homogenous manner by the other cells. Is there another approach to test the re-uptake of these tagged agents and whether secondary drug exposure is a persistent issue?

Third, the authors emphasize the importance of large cell numbers in order to minimize or suppress the variability in cell responsiveness; Can the authors provide some data to demonstrate distinct experiments with different numbers of cells?

*Reviewer #2 (Recommendations for the authors):*

Overall this is a very interesting concept and manuscript. More clarity is needed around the set up for the injections – perhaps pictures or diagrams would be helpful for readers early on to grasp what is occurring. Reading through the methods, they appeared more result-like in their writing; perhaps points that wish to be made in the methods could be pulled out and placed into the results.

Regarding the validity of the approach. What is being proposed is very novel for chemotherapeutic agents, but there is a working parallel that might be useful to prove the concept: that of multi-drug combinations in microbiology cultures. While that is beyond the scope of this paper, a similarly established quantitative multi-drug model with a quantifiable outcome would ideally have been used for determining the validity of your new methodology.

The only other recommendation I would make on an otherwise excellent paper is to clarify the usage of microRNA experimentation. The rationale for those experiments seemed disjointed from the combo chemotherapy experiments. I would reword the manuscript to better set up the readers to understand the need for those experiments and how they added to the validity of the model.

*Reviewer #3 (Recommendations for the authors):*

In addition to the comments listed in the Public Review, I have more detailed comments on the manuscript to improve the scientific communication:

Advection vs convection: terminology seems to be misused several times in the text

"Barcoding": this terminology is highly misleading, especially since single-cell heterogeneity is often explored through oligonucleotide barcoding (see PMID: 33619394).

Methods: What were the parameters used in microscopy imaging to prevent PMT saturation? How is the threshold for minimal detection handled in linear regression?

Methods: What buffer fluid was used with the labeling kits? In the homogenous concentration setup, were cells in a density-matched fluid?

Nearly all of the figures require more detailed schematics to indicate how (and in what order) the dye(s) or drug(s) are introduced into the cuvette or culture dish

Figures 1d and 2b: The correlation matrices are not symmetrical. Why? Does this reflect the sequence of uptake? If point injection, wouldn't this be simultaneous delivery? Please provide more detail for interpretation.

Figure 3: Please provide R2 values for each linear regression. Figure 3f: It would be helpful to visualize a 3rd concentration (e.g. 15 uM) so that if there is in fact a different slope for homogenous concentrations that this could be further evaluated.

Figure 4c: Is this additive or synergistic repression? How does your pipeline enable distinguishing these scenarios?

Figure 7: No statistical analysis was provided with these graphs. What is the standard error of these measurements, considering that propagation of error associated with the normalization will be needed?

Figure 8: How is one to interpret the "pockets" of potency identified by the drug distributions? Does this correspond to known additivity or synergy with these particular drugs? At first glance, it appears that the experimental setup for point injections did not facilitate enough cells encountering intermediate concentrations of both drugs.

Figure 9: No color bar is provided for the figure. As mentioned above, absolute concentrations for the 4 drugs are needed here to demonstrate the utility of the approach. I cannot tell the gradients of each channel from the top figure other than the region of max fluorescence

Figure 10: Again, schematics of the delivery methods are needed for dish vs cuvette

Figure 11: I found this figure unhelpful and would prefer to have more detailed information on the spatial partitioning used in figure 9.

---

## [Author Response]

Reviewer #1 (Recommendations for the authors):In this paper, the authors develop a method that allows one to test a large number of drug combinations in a single cell culture sample. Their new work is based on their previous studies with drug targeting at the single-cell level. In principle, the experiments rely on randomness of drug uptake in individual cells as a tool to create and encode drug treatments, and for the most part, the experiments validate that approach. The siRNA studies are particularly provocative and provide a novel means of using combinatorial si and i-RNA agents. As such this manuscript has the potential to change how investigators test combinatorial treatments with a single-cell approach. Nevertheless, there are several concerns that should be addressed through other experiments or further discussion of the limitations beyond the issue of random mixing of drugs.First, did the authors use other cell lines to see if their approach is applicable, especially in non-malignant primary cells or distinct cell lines.

We tested random dye combinations in four human cell lines: HeLa, HEK293 (tumorigenic embryonic kidney cells), MDA-MB-231 (epithelial breast cancer cells), and HPAEC (pulmonary artery endothelial cells). Functional experiments were conducted only using HeLa and HPAEC (nonmalignant cell line); please refer to auranofin conjugation-free barcoding in the HPAEC line described in the section, "Conjugation-free Drug Barcoding," with the results shown in Figure 4. While the method should be applicable to any cell type, the drug barcoding (either using conjugation free or nanoparticle labeling) must be optimized in terms of concentrations and exposure time. Higher concentrations of label and longer exposure may result in toxicity for the agents we tested, and the quantitation of this response is specific to cell type, as well.

Second, much of the application success is based on the assumption that there is minimal leakage across cells of the fluorescent tags, and that whatever does leak is redistributed in a homogenous manner by the other cells. Is there another approach to test the re-uptake of these tagged agents and whether secondary drug exposure is a persistent issue?

We thank the referee for this important, critical comment. In response, we added a paragraph in the Methods section addressing this issue (line 542). Leakage can occur on short and long time scales, based on the following reasoning.

Short time scale leakage occurs during the wash/collection steps using media (containing both fluorescent tags and drugs) when sample is unavoidably homogenized. This phenomenon, in turn, can affect local labeled drug gradients in culture, reducing the dynamic range of drug concentrations within the sample. Cells that were initially exposed to very low local drug(s) concentration are most affected. Thus, the washing steps need be performed quickly using large dilution volumes. We note that this short scale leakage effect occurs with both label(s) and drug(s) proportionally. Hence, label intensity should still be proportional to the corresponding drug concentration, and can be used as a proxy for drug concentration.

Long time scale leakage occurs if the drug(s) within cells is released to neighboring cells either via media or through cell-to-cell contact. In general, this secondary drug exposure cannot be accounted for using fluorescent tags in our method because the tags/labels are designed to be irreversibly bound to the cells and do not leak (labels can, however, be diluted by cell division). Drug leakage from cell to cell mediated by media can be estimated using the volume ratio of cells and media. This ratio in most cell culture conditions is negligibly small. For example, 10^4^ HeLa cells have a volume of ∼2.5⋅10−2μl (https://bionumbers.hms.harvard.edu/bionumber.aspx?id=103725&ver=20). Hence, in a well that contain 250μl of media, this ratio is ∼10−4.

Drug leakage from cell to cell mediated by direct contact can be much more significant. However, this effect is mitigated in adherent cell culture experiments by the coarse-graining procedure we use when the optical field containing multiple neighboring cells is averaged.

Third, the authors emphasize the importance of large cell numbers in order to minimize or suppress the variability in cell responsiveness; Can the authors provide some data to demonstrate distinct experiments with different numbers of cells?

We discuss this issue in the Supplementary Material section, Noise and Law of Large Numbers (please see Figure S1 in particular). In brief, our method relies on sub-population averaging to suppress noise in cell response due to inherited cell variability. This averaging (coarse-graining) procedure reduces intrinsic noise by factor r, where r is the number of cells in the sub-population. The typical coefficient of variation, CV, in biological applications is less than 1, and averaging over 100 cells reduces the intrinsic noise to CV∼0.1. Further noise reduction is probably meaningless owing to the existence of other factors such as systematic instrumental noise. Moreover, averaging over too many cells results in shrinking the dynamic range of variable(s) that encode drug concentration and cell phenotype. In the extreme case of averaging over an entire cell population, one can only determine a single data point, reducing the ‘single’ cell to bulk measurement.

In order to investigate the accuracy of prediction as a function of cell number in the population, we simply down-sampled (randomly) existing experimental data to reduce the number of cells in the ensuing *in silico* experiments. (Please see the Supplementary Material section, Noise and Law of Large Numbers, for more details.) We believe that this approach is equivalent to performing distinct real experiments with different numbers of cells. (We wish to emphasize that physical experiments with different numbers of cells would need to be conducted using scaled volumes to ensure identical cell/drug concentrations, including local injection volumes, which could be fairly challenging experimentally.)

Reviewer #2 (Recommendations for the authors):Overall this is a very interesting concept and manuscript. More clarity is needed around the set up for the injections – perhaps pictures or diagrams would be helpful for readers early on to grasp what is occurring. Reading through the methods, they appeared more result-like in their writing; perhaps points that wish to be made in the methods could be pulled out and placed into the results.

We thank the referee for this suggestion. In response to this comment, we reorganized the paper to move the discussion related to creation of local gradients and the coarse-graining procedure into the Results section. We also created a graphic abstract (listed as Figure 1 in the revised manuscript) that we hope to be useful for readers early on.

Regarding the validity of the approach. What is being proposed is very novel for chemotherapeutic agents, but there is a working parallel that might be useful to prove the concept: that of multi-drug combinations in microbiology cultures. While that is beyond the scope of this paper, a similarly established quantitative multi-drug model with a quantifiable outcome would ideally have been used for determining the validity of your new methodology.

We thank the reviewer for this very insightful observation. We note that the method, "disc-diffusion antibiotic susceptibility (Kirby–Bauer) test," used in microbiology cultures was the inspiration for this project in the first place. We tried to imagine what would happen to local microbial density if the diffusion disks of two different drugs would overlap. The schematic depiction shown in Figure 12 was our initial view of this scenario. The reviewer’s suggested application in microbial cultures is, indeed, an excellent example for testing drug combinations and, perhaps, is easier to implement compared to mammalian cell cultures. We are very grateful for this suggestion.

The only other recommendation I would make on an otherwise excellent paper is to clarify the usage of microRNA experimentation. The rationale for those experiments seemed disjointed from the combo chemotherapy experiments. I would reword the manuscript to better set up the readers to understand the need for those experiments and how they added to the validity of the model.

One of the motivations for the inclusion of two entirely different biological topics in the same paper was to demonstrate a range of possible applications of our method using different drug classes. Another motivation in considering the siRNA application was a simple phenotype readout. We assessed the effect of a combination of siRNAs on a direct and quantifiable target, viz., GFP fluorescence, as modified by siRNA-dependent expression of GFP protein. Unlike combination chemotherapy experiments, this is a much better understood and simpler system. In response to the reviewer’s suggestion, we added a linking paragraph in the Introduction’s text for clarification (line 112).

Reviewer #3 (Recommendations for the authors):I have more detailed comments on the manuscript to improve the scientific communication:Advection vs convection: terminology seems to be misused several times in the text

The reviewer is correct. We changed the terminology in the manuscript in response to this comment using the following rule – we referred to dispersion of dye/drug by local fluid injection as an advection process. The movement of fluid/air due to thermal and other gradients around the sample is referred to as convection.

"Barcoding": this terminology is highly misleading, especially since single-cell heterogeneity is often explored through oligonucleotide barcoding (see PMID: 33619394).

We agree that "barcoding" terminology in biology usually implies genetic barcoding. Here, however, we use this term much as Gary Nolan’s group had over a decade ago (cf., Krutzik PO, Nolan GP. Fluorescent Cell Barcoding in Flow Cytometry Allows High-Throughput Drug Screening and Signaling Profiling., Nature Methods, 2006). We added clarification on this matter in the "Introduction" section (line 83).

Methods: What were the parameters used in microscopy imaging to prevent PMT saturation? How is the threshold for minimal detection handled in linear regression?

The reviewer’s question is related to the fact that to image many cells using scanning microscopy, one has to image and stitch/combine many optical fields together. Since we utilize gradient labeling, some cells may have very high signal intensities, while other cells in the population may have very low fluorescent signal intensities–below the detection limit for a given PMT voltage controlling detector’s sensitivity. Additionally, if measurement involves multiple samples (e.g., gradient and homogeneously labeled specimens), the dynamic range of detector sensitivity has to accommodate a broad range of possible fluorescent signals.

We, therefore, used the following strategies to avoid signal saturation:

All related samples were imaged using the same PMT settings for each individual channel. This means that the imaging was not optimal for some conditions but allows one to compare cell label intensities directly.We scanned all related samples manually (using a wide-field camera and eye-piece) to identify the brightest (highest intensity) sub-population. (This can be done for fixed samples only.) The maximal PMT voltage setting for each channel was set to avoid signal saturation for these brightest spots (their location can vary across sample(s) for each respective channel).

We used Cellpose v0.6 for morphological/texture segmentation using brightfield images (with default values for flow and cell threshold). This implementation was done to make image segmentation independent of fluorescent tag signal(s) and avoid dealing with fluorescent threshold detection differences that can vary across tiles and samples.

For low signal intensity handling, we averaged the signal over a cell cross-section of at least 300 pixels, minimizing white (but not systematic) instrument noise. Background (white noise) signal noise was assessed for each optical field using inversion of the binary (mask) of foreground objects identified by Cellpose.

Methods: What buffer fluid was used with the labeling kits? In the homogenous concentration setup, were cells in a density-matched fluid?

Serum-free media and dPBS were used as buffers with the labeling kits. We did not match media density to cell density in the nonhomogeneous setup to minimize movement of the cells during injection. To ensure similar conditions, we also did not match cell and media densities in the homogeneous setup, as well. We clarified this point in Methods (line 509).

Nearly all of the figures require more detailed schematics to indicate how (and in what order) the dye(s) or drug(s) are introduced into the cuvette or culture dish

In response to this comment, we listed details of the dye/drug injection method in the Methods section (line 500) and modified the corresponding figure captions. In brief, suspension/detached cells were treated one dye at a time to create independent gradients within a sample. The only exception was a co-delivery experiment in which a pair of the pre-mixed dyes was injected simultaneously. In all experiments with adherent cell culture, drugs/dyes were injected into different slide locations simultaneously.

Figures 1d and 2b: The correlation matrices are not symmetrical. Why? Does this reflect the sequence of uptake? If point injection, wouldn't this be simultaneous delivery? Please provide more detail for interpretation.

We thank reviewer for this clarifying question. The reviewer correctly states that the switch in delivery sequence should not result in a dramatic change in dye-dye correlation coefficients.

The asymmetry is not due to the sequence of delivery. The lower and upper triangular parts of the tables in Figures 1d and 2b simply represent different delivery methods as stated in the legend: "Correlation coefficients, ρ, for all pairwise dye combinations are shown graphically for homogeneous and point injection deliveries on lower and upper triangular matrix plots, respectively."

Perhaps, the culprit for the confusion was a shared diagonal (self-correlation coefficient is 1 regardless of the delivery method). We adjusted the legend to stress the distinction between upper and lower diagonal parts of the correlation tables, and to clarify the confusing prior version.

Figure 3: Please provide R2 values for each linear regression. Figure 3f: It would be helpful to visualize a 3rd concentration (e.g. 15 uM) so that if there is in fact a different slope for homogenous concentrations that this could be further evaluated.

In response to this comment, we added adjusted-R2 values for linear regressions in Figure 3. We also performed additional homogeneous staining experiments using a 15μM dye concentration as the reviewer suggested. We used significantly higher cell numbers (5 fold) compared to previous experiments to define better the slope (present due to variability in dye uptake even in homogeneous staining conditions). The results are shown in updated Figure 3f.

Figure 4c: Is this additive or synergistic repression? How does your pipeline enable distinguishing these scenarios?

Throughout this paper, we focus exclusively on the measurement approach rather than on data interpretation, which is context- and phenotype-dependent. For example, the typical pharmacological use of a synergy/antagonism methodology is applicable exclusively to "exponential" systems, such as those measuring the effect of drug combinations on cell density/count. The exponent is the only function that satisfies the condition, f(x+y)=f(x)f(y)=f(y)f(x), which renders the effect of the combination tractable using standard methods, such as Bliss Independence, Loewe Additivity, etc. (Here f(x, y) is the observed phenotype as a function of concentrations x, y of two different drugs.) Both processes of cell death and division occur proportionally to the number of cells in the culture, and modulation of either of these processes would have an exponential effect on cell count. (Of course, this is true only in the exponential growth phase.)

The miRNA effect on gene expression is not, in general, a first order process. The modulation of expression occurs by means not only of accelerated degradation of target mRNA (which is, indeed, an exponential decay process), but also by modulation of translation. The translation modulation is not an exponential process because it does not change the number of mRNAs and affecting a different species – proteins. Hence, quantification of synergy may require more complicated analysis, which we did not address in this paper.

Figure 7: No statistical analysis was provided with these graphs. What is the standard error of these measurements, considering that propagation of error associated with the normalization will be needed?

In response to this comment, we included error bars in both Figure 7a and 7b. We note that while homogeneous drug delivery in biological samples can be replicated in traditional fashion, local delivery samples are not reproducible in our current implementation. We intentionally exploit randomness in the gradient sample preparation and, thus, no two samples are alike (please also see response to your comment above). Hence, the starting drug mixing conditions and subsequent sample cell population response are decisively unique.

One has to isolate cell subpopulations with a specific drug combination in order to be able to deduce the fidelity of the phenotype for multiple gradient samples. We have not performed this analysis here. Instead, the analysis shown in Figure 7b represent the average over the *entire* cell population, and hence, the result(s) is also unique for each sample.

However, it is possible to estimate the standard error of technical repeats. To this end, the entire gradient sample data were randomly reshuffled and partitioned into 3 "replicates." We averaged intensity data in each of these replicates and calculated the standard error.

Finally, the compound (due to normalization) standard error in Figure 7b was calculated as follows:

Var(x/y)≈(E(x)E(y))2(Var(x)E(x)2+Var(y)E(y)2)

Figure 8: How is one to interpret the "pockets" of potency identified by the drug distributions? Does this correspond to known additivity or synergy with these particular drugs? At first glance, it appears that the experimental setup for point injections did not facilitate enough cells encountering intermediate concentrations of both drugs.

Please see our response to your comments (8) regarding data interpretation. The docetaxel (TXT) and cyclophosphamide (CP) combination was synergistic against MA 13/C mammary adenocarcinoma cells in some preclinical studies (please see PubMed 19204201 and 9364540, for example). Only a few dose regimens were tested in these publications. We are not aware of extensive concentration/dosage testing of the combinations conducted in cell lines.

The reviewer is correct: stochastic drug mixing was not ideal and does not represent uniformly all possible conditions. However, it is still possible to infer drug combination action by comparing different time points in the data. Ideally, one has to sample drug distributions at time t=0 by collection of sub-samples, which is straightforward with suspension cell culture. Unfortunately, since we considered here adherent cells, this would require cell detachment and subsequent re-seeding of remaining cells back on the dish. This procedure was not well tolerated by drug-treated cells, and, as a consequence, we did not perform t=0 drug mixing normalization.

Figure 9: No color bar is provided for the figure. As mentioned above, absolute concentrations for the 4 drugs are needed here to demonstrate the utility of the approach. I cannot tell the gradients of each channel from the top figure other than the region of max fluorescence

We added a color bar to Figure 9 showing cell density levels for panels 2, 3, and 4. We also added a vertical panel showing smooth density distributions for each of the possible pairs of drugs (6). We also added a figure (Figure S16) and text (page 8) in the Supplementary Materials showing smooth contour (elevation) plots of estimated drug densities for each of the drugs.

Figure 10: Again, schematics of the delivery methods are needed for dish vs cuvette

We added a few paragraphs in the Methods section detailing the delivery steps needed for dish and cuvette systems, respectively (cf., pages 11-12).

Figure 11: I found this figure unhelpful and would prefer to have more detailed information on the spatial partitioning used in figure 9.

We hope that modifications of Figures 9 in response to reviewer comments 11 and 12 will make the presentation clearer. We also hope that the graphic abstract (Graphic Abstract Figure) can provide a clear schematic of the delivery strategy we used.

We believe that Figure 11 is useful to demonstrate that specific mixing conditions are determined by relative locations of injection points and can be used to identify stoichiometric local drug combinations. The finite sized intersection regions are helpful to justify the coarse-grain strategy used in this work.